# CALIBRATION IS GROUPING: VR-SAG WITH INTRA-GROUP VARIANCE CONTROL AND LOGIT-CLUSTER EVALUATION

## ABSTRACT

Accurate click-through and conversion-rate estimates are pivotal for bid optimization in large-scale advertising, yet modern deep CTR/CVR models are often miscalibrated. Classical global calibrators (Platt scaling, isotonic regression) and feature-based binning struggle to capture latent user–item heterogeneity. We approach calibration through the lens of *latent, calibration-aware groupings* and propose **Variance-Reduced Semantic-Aware Grouping (VR-SAG)**—a lightweight post-hoc layer over a frozen backbone that (i) forms semantically coherent partitions in embedding space, (ii) fits per-group temperature+bias calibrators, and (iii) explicitly penalizes intra-group variance to tighten probability spreads. Our design is grounded in a group-wise decomposition of proper scoring rules (e.g., Brier), which isolates intra-group variance as a key driver of residual miscalibration and motivates variance control for genuine loss reduction. To decouple evaluation from training, we introduce **Logit-Cluster Calibration Error (LCCE)**, an unsupervised fixed-partition metric obtained via $K$-means in logit space; LCCE aligns with the reliability term of proper scores while avoiding pitfalls of trainable grouping heads used as metrics. Across large-scale offline logs and **AdAuction**—a large-scale ad-auction dataset with oracle CTRs generated by an internal ad-auction simulator—VR-SAG consistently improves calibration (ECE/LCCE and Brier variants) over strong baselines, with negligible latency and memory overhead.

## 1 INTRODUCTION

Machine learning recommender systems underpin virtually every modern advertising platform, orchestrating the selection and pricing of *tens of billions* of ad impressions each day (Covington et al., 2016; Zhang et al., 2014b). For each impression, the model reports two probabilities—click-through rate (CTR) and conversion rate (CVR)—whose precision is crucial for both platform revenue and advertiser return (Richardson et al., 2007; He et al., 2014). Because an auction bid equals an advertiser's private value times one of these predicted probabilities, even modest calibration errors propagate into mispriced traffic and distorted budget pacing (McMahan et al., 2013). To satisfy strict latency and scale requirements, production systems typically employ deep architectures such as Wide & Deep (Cheng et al., 2016) and DeepFM (Guo et al., 2017b). While we motivate both CTR and CVR at a high level, all experiments in this paper focus on impression-level CTR calibration over exposure logs; extending the same ideas to CVR under stronger selection bias is left as future work.

Despite strong ranking performance, these models often produce *miscalibrated* probabilities: after grouping predictions into narrow bins, the observed click frequency rarely matches the average score. Such miscalibration erodes auction efficiency and reduces revenue (Lin et al., 2024), motivating extensive work on calibration for ads ranking (McMahan & Muralidharan, 2012; Fan et al., 2023; Borisov et al., 2018; Chaudhuri et al., 2017; Sheng et al., 2023). Existing approaches either learn a single global mapping (e.g., Platt scaling (Platt et al., 1999), isotonic regression (Zadrozny & Elkan, 2002; Niculescu-Mizil & Caruana, 2005), temperature scaling (Guo et al., 2017a))—which can leave significant residual error within subpopulations—or rely on predefined metadata partitions

(multi-calibration (Hébert-Johnson et al., 2018), field-aware methods (Pleiss et al., 2017)), which cannot capture latent behavioral regimes and may mask opposite biases within the same group.

To address these gaps, we adapt Semantic-Aware Grouping (SAG) (Yang et al., 2023) for CTR/CVR calibration and introduce three contributions. First, we derive a group-wise Brier-loss decomposition that reveals a variance-driven miscalibration term, and propose *Variance-Reduced SAG* (VR-SAG), which jointly learns per-group temperatures and biases while penalizing intra-group variance to enforce tighter, more coherent partitions. Second, we decouple evaluation from the trainable grouping head by defining the *Logit-Cluster Calibration Error* (LCCE), an unsupervised, fixed-partition metric in logit space that aligns with the reliability term of proper scoring rules. Finally, we construct **AdAuction**, a large-scale ad-auction dataset with oracle CTRs generated by an internal simulator that mirrors industrial auction workflows. In summary:

- We introduce a principled group-wise Brier-loss decomposition and leverage it to design VR-SAG, which combines semantic grouping with intra-group variance regularization for superior calibration under production constraints.

- We propose LCCE, a low-variance logit-space clustering metric that provides a better assessment of calibration quality while avoiding the pitfalls of trainable grouping metrics.

- We release **AdAuction**, a realistic ad-auction dataset with oracle CTRs generated by an internal simulator, facilitating rigorous and reproducible benchmarking of calibration techniques in large-scale recommender systems.

## 2 METHOD

We begin by reviewing binary-CTR prediction and the Expected Calibration Error (Sec. 2.1), then introduce Semantic-Aware Grouping (SAG), which applies group-specific temperatures over a frozen backbone (Sec. 2.2). Building on SAG, we derive a group-wise Brier-loss decomposition that isolates a variance-driven miscalibration term (Sec. 2.3) and propose variance-regularized VR-SAG to address it (Sec. 2.4). Finally, we present the Logit-Cluster Calibration Error (LCCE), an unsupervised, fixed-partition metric for calibration evaluation (Sec. 2.5).

### 2.1 BACKGROUND

Let an impression be represented by a feature vector $x \in \mathbb{R}^d$ obtained by concatenating user descriptors, ad metadata and real-time context, and let the click indicator be $y \in \{0, 1\}$ ($y = 1$ means the user clicked the ad). A predictor $f_\theta : \mathbb{R}^d \to [0, 1]$ parameterized by $\theta$ outputs the raw click-through probability $\hat{p} = f_\theta(x)$. Its penultimate layer produces a hidden representation $z(x) \in \mathbb{R}^m$, and its last linear layer returns a single logit $o(x) \in \mathbb{R}$ before the final sigmoid activation. The network is trained by minimizing the average negative log-likelihood

$$\mathcal{L}_{\text{CE}}(\theta) = -\frac{1}{n} \sum_{i=1}^{n} \Big[ y_i \log \hat{p}_i + (1 - y_i) \log(1 - \hat{p}_i) \Big], \tag{1}$$

a proper scoring rule(Gneiting & Raftery, 2007) that enforces accuracy but not probability calibration; modern CTR systems therefore remain miscalibrated, especially on sparse ad or user slices.

A predictor is *well-calibrated* when the conditional click frequency equals its score, i.e. $\Pr(Y = 1 \mid \tilde{p} = q) = q$ for all $q \in [0, 1]$. Practitioners monitor calibration with the Expected Calibration Error (ECE), the weighted average gap between predicted probability and empirical click rate across probability bins. Formally, partition $[0, 1]$ into $M$ equal-width bins $B_1, \ldots, B_M$; then

$$\text{ECE} = \sum_{m=1}^{M} \frac{|B_m|}{n} \big| \underbrace{\frac{1}{|B_m|} \sum_{i \in B_m} y_i}_{\text{acc}(B_m)} - \underbrace{\frac{1}{|B_m|} \sum_{i \in B_m} \hat{p}_i}_{\text{conf}(B_m)} \big|, \tag{2}$$

with lower values indicating better alignment between predicted probabilities and observed outcomes.

## 2.2 SEMANTIC-AWARE GROUPING FOR CTR CALIBRATION

Semantic-Aware Grouping (SAG)(Yang et al., 2023) augments the frozen backbone with a lightweight grouping head. A weight matrix $W \in \mathbb{R}^{m \times K}$ and bias $b \in \mathbb{R}^K$ transform the embedding $z(x)$ into soft group weights

$$q_k(x) = \mathrm{softmax}\big(z(x)^\top W + b\big)_k, \quad k = 1, \ldots, K, \tag{3}$$

where $K$ is the chosen number of latent regions in embedding space that are expected to share similar calibration behaviour, rather than necessarily corresponding to single human-interpretable fields. We denote this softmax-based grouping head by $g_\phi$, with parameters $\phi = (W, b)$, so that $g_\phi(x) = q(x) = (q_1(x), \ldots, q_K(x))$. For each group we keep a single temperature $\tau_k > 0$. The calibrated click-through probability (binary) is then the mixture

$$\tilde{p}(x) = \sum_{k=1}^K q_k(x)\, \sigma\big(o(x)/\tau_k\big), \tag{4}$$

where $\sigma(t) = (1 + e^{-t})^{-1}$.

All added parameters $\phi = (W, b)$ and $\{\tau_k\}_{k=1}^K$ are learned jointly on a held-out validation set $D_{\mathrm{val}}$ with the SAG objective

$$\mathcal{L}_{\mathrm{SAG}} = -\frac{1}{|D_{\mathrm{val}}|} \sum_{(x,y) \in D_{\mathrm{val}}} \log\Big[\sum_{k=1}^K q_k(x)\left(y\,\tilde{p}_k(x) + (1-y)\big(1 - \tilde{p}_k(x)\big)\right)\Big] + \lambda \|W\|_2^2, \tag{5}$$

where $\lambda > 0$ regularizes the grouping weights, and $\tilde{p}_k(x) = \sigma\big(o(x)/\tau_k\big)$. We retain the soft weights $q_k(x)$ during both calibration and serving, avoiding hard arg-max reassignment. At inference the extra cost is one $m \times K$ matrix–vector product and $K$ scalar operations, negligible compared with the backbone forward pass.

**Why it helps in production.** Soft semantic partitions let each temperature specialize to coherent behavioral regimes—user cohorts, ad creatives, time-of-day effects—while still letting tail impressions borrow strength from related high-volume traffic, a property single-temperature or binning methods lack. In our usage, "semantic" therefore refers to calibration-aware structure already encoded in $z(x)$ by the CTR/CVR backbone, not to perfectly disentangled or manually labelled concepts.

## 2.3 DECOMPOSITION OF PROPER SCORING RULES WITH SEMANTIC GROUPS

Probabilistic models are ideally evaluated using *proper scoring rules* (Gneiting & Raftery, 2007), i.e., loss functions whose expected value is uniquely minimized when the predicted probabilities coincide with the true data-generating probabilities. In our binary CTR setting, both negative log-likelihood and the Brier score are proper, which motivates using them as the main training and analysis tools in what follows. Popular calibration metrics such as ECE, while intuitive, lack this property and can be gamed without improving true predictive fidelity. To bridge this gap we expose how calibration terms reappear inside a proper scoring rule once predictions are partitioned into semantic groups. For clarity we detail the case of the *Brier score*; the same reasoning carries over to other proper scoring rules—including cross-entropy— yielding analogous insights with different algebraic constants[1]. The decomposition that follows clarifies when reducing a calibration error genuinely lowers a proper loss and when it merely provides a misleading signal.

Let $G_k$ be the $k^{\mathrm{th}}$ latent region induced by $g_\phi$ and denote

$$w_k = \Pr(G_k), \quad \pi_k = \Pr(Y = 1 \mid G_k), \quad \mu_k = \mathbb{E}[\hat{p} \mid G_k],$$

where $\hat{p} = f_\theta(x)$ is the *uncalibrated* probability output of the frozen backbone. Write the within–group variance $\sigma_k^2 = \mathrm{Var}(\hat{p} \mid G_k)$ and covariance $\gamma_k = \mathrm{Cov}(\hat{p}, Y \mid G_k)$.

For binary events the Brier loss is $S(Y, \hat{p}) = (Y - \hat{p})^2$. Conditioning on $G_k$ and using $Y^2 = Y$ gives

$$\mathbb{E}[S \mid G_k] = \pi_k - 2\,\mathbb{E}[\hat{p}Y \mid G_k] + \mathbb{E}[\hat{p}^2 \mid G_k].$$

---

[1]See the Appendix for detailed proofs and analysis of other proper scoring rules.

Because $\mathbb{E}[\hat{p}Y \mid G_k] = \mu_k \pi_k + \gamma_k$ and $\mathbb{E}[\hat{p}^2 \mid G_k] = \mu_k^2 + \sigma_k^2$, we obtain

$$\mathbb{E}\big[(Y - \hat{p})^2 \mid G_k\big] = \pi_k(1 - \pi_k) + (\pi_k - \mu_k)^2 + \sigma_k^2 - 2\gamma_k.$$

Averaging over groups yields the **grouping decomposition**

$$\mathbb{E}[S] = \underbrace{\bar{Y}(1 - \bar{Y})}_{\text{UNC}} + \underbrace{\sum_k w_k(\mu_k - \pi_k)^2}_{\text{REL}} - \underbrace{\text{Var}(\pi_k)}_{\text{RES}} + \underbrace{\sum_k w_k(\sigma_k^2 - 2\gamma_k)}_{\Delta}, \qquad (6)$$

where $\bar{Y} = \mathbb{E}[Y]$ denotes the marginal click rate. Using the law of total variance, $\sum_k w_k \pi_k(1 - \pi_k) = \bar{Y}(1 - \bar{Y}) - \text{Var}(\pi_k)$. The classical UNC + REL − RES(Murphy, 1973) form is recovered only when every group collapses to a single forecast value so that $\sigma_k^2 = \gamma_k = 0$.

Because SAG's soft regions preserve a spread of scores ($\sigma_k^2 > 0$) and the sign/magnitude of $\gamma_k$ varies across datasets, the extra term $\Delta$ is often positive in practice and increases the Brier loss. More generally, $\Delta$ is not sign-definite (see Appendix): each contribution $w_k(\sigma_k^2 - 2\gamma_k)$ can be positive or negative depending on the variance–covariance balance, but for fixed $\gamma_k$ it is monotone in $\sigma_k^2$, so reducing within-group variance always decreases that group's contribution to $\mathbb{E}[S]$ regardless of the sign pattern. Using a Cauchy–Schwarz bound (Appendix A.2), we have $|\gamma_k| \le \sigma_k \sqrt{\pi_k(1 - \pi_k)}$, which implies that $(\sigma_k^2 - 2\gamma_k)$ is typically dominated by $\sigma_k^2$ in sparse CTR regimes where $\pi_k$ is small; this makes within-group variance a stable and tractable surrogate for shrinking $\Delta$, whereas directly penalizing $\gamma_k$ would require estimating noisy covariances. VR-SAG counters this effect with a variance penalty $\lambda_v \sum_k w_k \sigma_k^2$, which contracts the spreads and empirically pulls predictions toward the local mean. As both quantities shrink, the overall reliability term—and therefore the expected Brier score—decreases, offering a principled explanation for the effectiveness of Variance-Reduced SAG that will be introduced in Sec. 2.4.

**Definition 2.1** (Grouping Calibration Error (GCE)). Given the latent regions $\{G_k\}_{k=1}^K$ induced by the grouping function $g_\phi$, let $w_k = \Pr(G_k)$, $\mu_k = \mathbb{E}[\hat{p} \mid G_k]$ and $\pi_k = \Pr(Y = 1 \mid G_k)$. The *Grouping Calibration Error* of a probabilistic predictor $f_\theta$ with respect to this partition is

$$\text{GCE}(g_\phi; f_\theta) = \sum_{k=1}^K w_k(\mu_k - \pi_k)^2. \qquad (7)$$

Equation 7 is identical to the **REL** term in the grouping decomposition of the Brier loss given in equation 6. Hence the choice of partition $\{G_k\}$ has a first–order impact on both the measured calibration error and its gap to any proper scoring rule that admits such a decomposition: partitions that bring $\mu_k$ closer to $\pi_k$ simultaneously reduce GCE and the overall scoring loss, providing a tighter assessment of probabilistic accuracy. In VR-SAG we therefore use the Brier decomposition and GCE as analytic tools: the actual training objective remains a grouped NLL plus a variance penalty rather than a direct minimization of GCE, which helps avoid degenerate partitions.

**Re-expressing classical calibration metrics via grouping.** The grouping perspective unifies several existing metrics:

- **Singleton groups.** When every impression forms its own group ($K = n$ and $G_k = \{(x_i, y_i)\}$), we have $w_k = \frac{1}{n}$, $\mu_k = \hat{p}_i$, and $\pi_k = y_i$. Substituting these quantities in equation 7 gives $\text{GCE} = \frac{1}{n} \sum_{i=1}^n (\hat{p}_i - y_i)^2$, *exactly the Brier score.*[2]

- **Probability-based binning.** If instances are grouped according to their predicted probability—for example into $M$ equal-width or equal-frequency bins—each bin $B_m$ acts as a region $G_k$. Then $\mu_k$ equals the bin's average confidence, $\pi_k$ equals its empirical accuracy, and GCE reduces to the weighted sum of squared (accuracy–confidence) gaps that underlies the squared-ECE variant.

These examples illustrate that *the partition is the metric*: a well-chosen, semantically meaningful grouping not only lowers GCE but also sharpens the link between calibration error and the underlying proper scoring rule, yielding a more faithful view of predictive reliability.

---

[2] With singleton groups the uncertainty and resolution terms in equation 6 vanish, so the Brier score coincides with the reliability component.

## 2.4 VARIANCE-REDUCED SEMANTIC-AWARE GROUPING (VR-SAG)

Let the validation set contain $n = |D_{\text{val}}|$ impressions indexed by $i = 1, \ldots, n$. For each impression $x_i$ the frozen backbone produces a raw score $\hat{p}_i = f_\theta(x_i)$, a hidden vector $z_i = z(x_i)$ and logit $o_i = o(x_i)$. The *grouping head* $g_\phi$ (parameters $\phi = (W, b)$) returns soft assignments

$$q_{ik} = \text{softmax}(z_i^\top W + b)_k \quad k = 1, \ldots, K,$$

to $K$ latent regions $\{G_k\}$. For each group we keep a temperature $\tau_k > 0$ and a bias $\beta_k \in \mathbb{R}$, so that the calibrated probability is

$$\tilde{p}_i = \sum_{k=1}^{K} q_{ik} \, \sigma\big(o_i/\tau_k + \beta_k\big). \tag{8}$$

This amounts to per-group Platt scaling on the backbone logits, mixed by the soft semantic assignments $q_{ik}$. When all $\beta_k = 0$ this reduces to temperature scaling; learning both $\{\tau_k\}$ and $\{\beta_k\}$ recovers per-group Platt scaling.

We define three empirical statistics per group:

$$\hat{w}_k = \frac{1}{n} \sum_{i=1}^{n} q_{ik}, \quad \hat{\mu}_k = \frac{1}{n\hat{w}_k} \sum_{i=1}^{n} q_{ik} \, \hat{p}_i, \quad \hat{\sigma}_k^2 = \frac{1}{n\hat{w}_k} \sum_{i=1}^{n} q_{ik} \big(\hat{p}_i - \hat{\mu}_k\big)^2, \tag{9}$$

i.e., the *normalized soft mass*, the mean uncalibrated score, and the within-group variance, respectively. These hatted quantities are standard plug-in estimators of the population-level $w_k, \mu_k, \sigma_k^2$ that appear in the grouping decomposition and are consistent under the usual i.i.d. sampling assumption.

**Variance reduction.** As shown in the grouping decomposition (equation 6), the mixture-of-temperatures-and-biases estimator in equation 8 incurs an extra term $\Delta = \sum_k w_k(\sigma_k^2 - 2\gamma_k)$. Reducing the intra-group variance $\sigma_k^2$ thus tightens an upper bound on the Brier score. We achieve this via the penalty

$$\mathcal{L}_{\text{VAR}} = \lambda_v \sum_{k=1}^{K} \hat{w}_k \, \hat{\sigma}_k^2, \tag{10}$$

with tunable weight $\lambda_v > 0$. Here $\sigma_k^2 = \text{Var}(\hat{p} \mid G_k)$ is defined for the *uncalibrated* backbone probabilities $\hat{p} = f_\theta(x)$ in our decomposition; penalizing this quantity keeps the regularizer aligned with the analytic source of Brier loss and shapes the grouping head $g_\phi$ toward regions where backbone scores are homogeneous, while leaving the calibrated outputs $\tilde{p}_i$ free to adjust under the NLL term. Combined with soft assignments $q_{ik}$, this biases the learned groups toward calibration-relevant, low-variance regions in embedding space and away from spurious noisy directions.

**VR-SAG objective.** Combining these elements, the validation-time loss is

$$\mathcal{L}_{\text{VR-SAG}} = -\frac{1}{n} \sum_{i=1}^{n} \log\Big[\sum_{k=1}^{K} q_{ik} \big(y_i \tilde{p}_{ik} + (1 - y_i)\big(1 - \tilde{p}_{ik})\big)\Big] + \lambda \|W\|_2^2 + \mathcal{L}_{\text{VAR}}$$

$$= \mathcal{L}_{\text{SAG-B}} + \lambda_v \sum_{k=1}^{K} \hat{w}_k \, \hat{\sigma}_k^2, \tag{11}$$

where $\tilde{p}_{ik} = \sigma(o_i/\tau_k + \beta_k)$ and $\mathcal{L}_{\text{SAG-B}}$ denotes the SAG objective in equation 5 extended to include per-group biases $\{\beta_k\}$.

Intuitively, the grouped NLL term $\mathcal{L}_{\text{SAG-B}}$ aligns per-group predictions with empirical click rates (acting on the REL/RES components of the proper-score decomposition), while the variance term $\mathcal{L}_{\text{VAR}}$ specifically shrinks the heterogeneity block $\Delta$ by reducing $\hat{\sigma}_k^2$; VR-SAG thus uses the decomposition as an analytic guide rather than optimizing UNC + REL − RES + $\Delta$ directly.

Minimizing $\hat{\sigma}_k^2$ via equation 10 contracts the extra term $\Delta$ in the decomposition, pulling predictions toward each group mean, typically lowering both covariance $\gamma_k$ and the calibration gap $|\pi_k - \mu_k|$, and yielding consistent empirical improvements in ECE. Because the backbone logits $o_i$ and their induced ranking are left unchanged and only passed through smooth, per-group Platt/temperature

transforms, these improvements are achieved with limited perturbation to the underlying score ordering (see Sec. E.1 for accuracy-preserving statistics).

VR-SAG retains all of SAG's production-friendly properties:

- No backbone retraining: only $\phi, \{\tau_k, \beta_k\}$ are updated.
- Minimal memory/latency cost: $K(m + 2)$ extra parameters and one $m \times K$ matrix–vector product per impression.
- Robustness and controllable impact on tail traffic: per-group temperatures and biases provide adaptive correction for under-represented slices, while the variance penalty (scaled by $\lambda_v$) suppresses noisy within-group spreads with only mild perturbation to the backbone ranking.

## 2.5 LOGIT-CLUSTER CALIBRATION ERROR (LCCE)

As introduced in Sec. 2.3, the Grouping Calibration Error in equation 7 measures the reliability term of a proper scoring rule under a partition $g_\phi$. While GCE benefits from data-adaptive partitions, its coupling to the trainable grouping head can mask true miscalibration by driving GCE down even when predictions remain poorly aligned with outcomes. To preserve the low-variance, model-aware slicing of GCE *without* a learned component, we define the *Logit-Cluster Calibration Error* by applying the same squared-gap measure to clusters formed in logit space.

Let the frozen backbone produce logits $o(x_i) \in \mathbb{R}$ and predicted probabilities $\hat{p}_i = \sigma\big(o(x_i)\big)$, where $\sigma(t) = (1 + e^{-t})^{-1}$. Perform $K$-means on $\{o(x_i)\}_{i=1}^n$ to obtain clusters $\mathcal{M} = \{M_j\}_{j=1}^K$. For each cluster $M_j$, define

$$w_j = \frac{|M_j|}{n}, \quad \mu_j = \frac{1}{|M_j|} \sum_{i \in M_j} \hat{p}_i, \quad \pi_j = \frac{1}{|M_j|} \sum_{i \in M_j} y_i.$$

The LCCE is then

$$\text{LCCE}_K = \sum_{j=1}^K w_j \left(\mu_j - \pi_j\right)^2, \tag{12}$$

which coincides with $\text{GCE}(g_{\text{logit}}; f_\theta)$ for the static, logit-based partition $g_{\text{logit}}$. By fixing the grouping, LCCE retains the variance advantages of clustering while avoiding the pathological minimization of GCE by a trainable head. Training thus proceeds with the learned semantic partition $g_\phi$ and the grouped NLL+variance objective, whereas LCCE is computed on a separate, fixed logit-based partition $g_{\text{logit}}$ that cannot be trivially optimized by changing $g_\phi$. In all experiments we use a small, fixed number of clusters (default $K = 4$); the sensitivity study in Sec. 3.2 and Appendix C shows that LCCE's model ranking is stable across a moderate range of $K$, so the exact choice is not critical as long as $K$ is kept modest.

**Why logits?** Clustering in logit space yields *model-aware Voronoi cells*: equal-sized intervals in $o$-space map to non-uniform bins in probability space, adapting to both the score distribution and the decision boundary. This prevents the extreme sparsity in probability tails seen with uniform binning, without relying on an optimized grouping head. Moreover, $K$-means is run on one-dimensional logits rather than high-dimensional embeddings, so cluster boundaries reduce to thresholds on the real line and are far less sensitive to different random initializations. In practice we average over 4 random $K$-means runs; details in Appendix C, which further damps seed-level noise and yields a low-variance, but not strictly deterministic, grouping metric that empirically tracks Brier-based proper scores more stably than alternatives such as ECE.

## 3 EXPERIMENT

We first conduct a comprehensive analysis of calibration error metrics. We then evaluate our method offline on two widely used public datasets—AliCCP (Ma et al., 2018) and AliExpress (Xu et al., 2019)—as well as on our newly open-sourced AdAuction dataset.

**Dataset with ground-truth CTR** We evaluate on **AdAuction**, an impression ad-auction dataset with oracle CTRs generated by an internal simulator. Like AuctionNet (Su et al., 2024), the underlying simulator retains the core workflow logic of industrial advertising systems—where auto-bidding agents process advertiser objectives, execute bid decisions, and collect post-auction feedback—thereby simulating inherent challenges such as sample selection bias (SSB). Specifically, its bidding mechanism mimics the natural overestimation issue: an overestimated ad item tends to win auctions more frequently and gains higher exposure during ranking, reflecting real-world biases in ad delivery.

Unlike AuctionNet, which solely records observable metrics (e.g., clicks/conversions), AdAuction incorporates ground-truth click-through rates as synthetic labels in its exposure data—an oracle signal inaccessible in real-world applications. This design enables direct calibration-error measurement against known truth values, a critical advantage for validating probabilistic prediction models that remains fundamentally unattainable in operational advertising platforms. The dataset contains 15M exposure samples with 451K clicks, and the AdAuction feature schema and labels are released for public use.

## 3.1 EXPERIMENTAL COMPARISON

Table 1: Comparison of calibration methods. Bold indicates statistically superior ($p < 0.05$) results. Here $\text{Brier}^+ = \mathbb{E}[\,|\pi - \hat{p}|\,]$ (MAE of probability error) and $\text{Brier} = \mathbb{E}[\,(\pi - \hat{p})^2\,]$. *Note:* On **AdAuction**, Brier and $\text{Brier}^+$ are computed against oracle CTR $\pi$; on **AliCCP** and **AE**, only ECE/LCCE are reported.

| | AdAuction | | | | AliCCP | | AE | |
|---|---|---|---|---|---|---|---|---|
| Method | ECE | LCCE | Brier$^+$ | Brier | ECE | LCCE | ECE | LCCE |
| Uncal | 0.0339 | 0.0342 | 0.0352 | 0.0527 | 0.2131 | 0.2161 | 0.2562 | 0.2562 |
| Histogram binning | 0.0049 | 0.0113 | 0.0170 | 0.0325 | 0.0185 | 0.0210 | 0.0168 | 0.0222 |
| Isotonic regression | 0.0056 | 0.0090 | 0.0123 | 0.0282 | 0.0076 | 0.0081 | 0.0079 | 0.0094 |
| Platt scaling | 0.0063 | 0.0124 | 0.0157 | 0.0297 | 0.0056 | 0.0071 | 0.0065 | 0.0077 |
| Temperature scaling | 0.0062 | 0.0124 | 0.0158 | 0.0296 | 0.0056 | 0.0059 | 0.0052 | 0.0061 |
| SAG+PS | 0.0058 | 0.0100 | 0.0131 | 0.0247 | 0.0027 | 0.0031 | 0.0027 | 0.0036 |
| SAG+TS | 0.0054 | 0.0099 | 0.0130 | 0.0249 | 0.0013 | 0.0017 | 0.0016 | 0.0020 |
| VR-SAG+PS(ours) | 0.0054 | **0.0083** | 0.0118 | **0.0237** | 0.0008 | 0.0008 | 0.0008 | 0.0010 |
| VR-SAG+TS(ours) | 0.0053 | **0.0083** | **0.0117** | 0.0239 | **0.0003** | **0.0005** | **0.0003** | **0.0005** |

We randomly partition a validation set $D_{\text{val}}$ from the standard training set (10% for validation), and reserve 10% of the standard test set as a hold-out calibration set $D_{\text{ho}}$. For each dataset–model combination, we perform 100 random test-set splits and report the average performance over 100 trials for each method. We conduct a paired $t$-test to assess statistical significance. Hyperparameters of comparative methods are tuned following their original papers, using 5-fold cross-validation. Unless otherwise noted, we fix the number of groups to 3 and the number of partitions to 10. The regularization strength is set to $\lambda = 0.1$, and the variance-penalty coefficient to $\lambda_v = 0.5$, following a similar tuning protocol as the baselines.

We compare the uncalibrated backbone (Uncal) against standard post-hoc calibrators and grouping-based methods: Histogram binning (Zadrozny & Elkan, 2001) (bin-wise averaging), Isotonic regression(Zadrozny & Elkan, 2002) (monotone piecewise mapping), Platt scaling (Platt et al., 1999) (logistic bias+scale), Temperature scaling (Guo et al., 2017a) (single temperature), SAG (Yang et al., 2023) (embedding-based semantic groups with per-group temperatures), and VR-SAG (ours) (SAG with intra-group variance control, evaluated with both PS/TS).

From Table 1, VR-SAG consistently outperforms the base calibrators on both ECE and LCCE, indicating improved calibration accuracy and strong generalization across datasets. Across all three datasets, the relative changes in accuracy and AUC between calibrated outputs and backbone logits are within roughly $\pm 0.005$ (see Appendix for the AccDiff/AUCDiff analysis), showing that VR-SAG improves calibration with negligible impact on ranking quality. Compared with SAG (Yang et al., 2023)—which shares a similar architecture but does not include the variance penalty—VR-SAG achieves further gains, highlighting the role of intra-group variance control. In contrast, classical binning-based calibrators such as histogram binning can noticeably reduce AUC by collapsing all

scores in a bin to the same value, while our smooth per-group Platt/temperature transformations perturb the ordering only mildly, leading to a more favorable calibration–ranking trade-off than coarse bin- or metadata-based post-hoc mappings.

### 3.2 ANALYSIS OF CALIBRATION-ERROR METRICS

We assess calibration metrics through a simple *rank-consistency* protocol: across many random train/validation/test splits and model variants, we compute each metric together with the oracle Brier score (available on AdAuction via ground-truth CTRs) and measure the Spearman rank correlation between the trial-wise rankings they induce. This abstracts away the absolute scale of a metric and asks only whether it orders models in the same way as a proper scoring rule.

As summarized in Figure 1, all metrics are positively correlated with the oracle Brier score, but grouping-based metrics are clearly more stable than pointwise ones, and LCCE achieves the highest Spearman correlation (greater than 0.9), consistently outperforming ECE and related alternatives. Detailed Monte Carlo settings, additional heatmaps, and second-order statistics (e.g., variance comparisons) are provided in Appendix C.

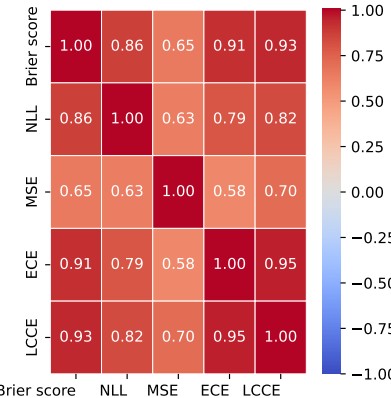

Figure 1: Spearman rank correlations between candidate calibration metrics and the oracle Brier score over many random trials. LCCE attains the strongest agreement with the proper score.

### 3.3 ABLATION STUDY

We conduct ablations on AdAuction to study three hyperparameters in our method: the number of partitions, the number of groups per partition, and the weight of the intra-group variance term. Results are summarized in Figure 2.

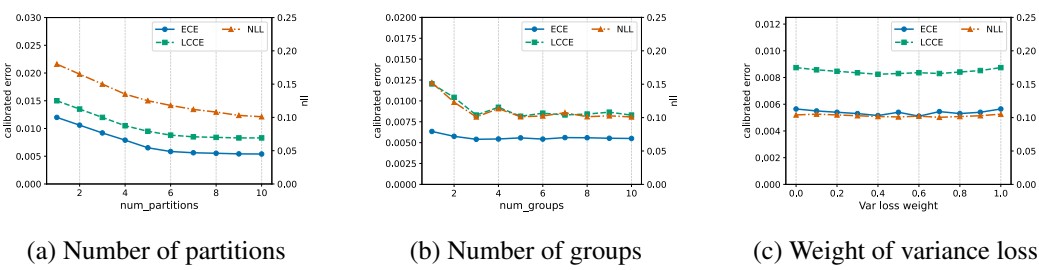

| (a) Number of partitions | (b) Number of groups | (c) Weight of variance loss |

Figure 2: Influence of key hyperparameters in VR-SAG (means over random seeds; no error bars shown).

As the number of partitions increases, calibration improves and becomes more stable, consistent with SAG (Yang et al., 2023); averaging across randomized runs reduces estimation noise. In contrast, increasing the number of groups per partition yields diminishing returns. This is expected from a bias–variance perspective: as groups proliferate, per-group sample support decreases, variance in the estimated temperatures/biases rises, and the net effect after mixing can cancel potential gains.

Moreover, with a frozen backbone and fixed feature budget, the effective heterogeneity captured by grouping saturates quickly; additional groups become highly correlated and do not provide new corrective directions for calibration. Regarding the variance-loss weight (Figure 2c), performance varies modestly across the tested range, suggesting partial complementarity rather than dominance.

We also visualize the group-wise distributions and confirm that VR-SAG yields substantially smaller within-group variance than field-based and plain SAG groupings (see Appendix H.3).

## 4 RELATED WORK

Early work on click-through-rate prediction treated it as supervised learning, from logistic regression for estimating click probabilities (Richardson et al., 2007) to large-scale online systems for industrial deployment (McMahan et al., 2013). In modern ad platforms, CTR/CVR models feed real-time bidding, where the revenue-optimal bid scales with the true response probability (Zhang et al., 2014a); reinforcement-learning agents further adapt bids for budgeted campaigns (Cai et al., 2017) but still depend on calibrated upstream probabilities.

Because perfect calibration is unattainable in finite samples, practitioners approximate it by binning predictions and comparing confidence with observed outcomes (de Menezes e Silva Filho et al., 2023). Expected Calibration Error (ECE) (Guo et al., 2017a) popularized this view and motivated simple post-hoc calibrators such as histogram binning (Chaudhuri et al., 2017), isotonic regression (Menon et al., 2012; Borisov et al., 2018), and Platt scaling (Platt et al., 1999). More recent "field-aware" approaches (Yang et al., 2024; Zhao et al., 2024) combine such mappings with user/item context and often reduce calibration error without degrading ranking metrics (Wei et al., 2022; Pan et al., 2020).

Field-aware calibration can be viewed as a specific instance of multi-calibration, which enforces calibration simultaneously over many (potentially overlapping) subpopulations (Hébert-Johnson et al., 2018). Beyond pre-specified partitions, a line of work learns the grouping itself using tree-based or data-driven partitioning schemes (Huang et al., 2022; Zadrozny & Elkan, 2001; Leathart et al., 2017; Durfee et al., 2022), though such groups can be cumbersome to integrate into deep recommender stacks and may optimize surrogate objectives only loosely aligned with probability calibration. Automatically discovered or latent regions (including ours) no longer coincide with simple field values such as *device* or *placement*, improving flexibility in capturing behavioral regimes but reducing direct human interpretability. Related ideas around grouping and calibration have also appeared in adjacent areas such as graph neural networks and confidence estimation for large language models (Seo et al., 2025; Zhuang et al., 2024; Detommaso et al., 2024), underscoring the broad interest in calibrated uncertainty across modern ML systems.

**Positioning.** Our approach differs in two key respects: (i) we freeze the backbone and learn *semantic* partitions directly in embedding space, equipping each group with its own temperature and bias; and (ii) we introduce variance regularization, motivated by a new grouping decomposition of proper scores that explicitly isolates the contribution of within-group variance. Unlike metadata- or rule-based partitioning, VR-SAG adapts to latent user–item regimes while imposing minimal serving overhead. For evaluation, LCCE fixes clusters in logit space, aligning with the reliability term of proper scoring rules and avoiding the pitfalls that arise when trainable groupings double as metrics.

## 5 CONCLUSION

We presented *Variance-Reduced Semantic-Aware Grouping* (VR-SAG), a lightweight post-hoc layer that calibrates CTR/CVR predictors by learning group-specific temperatures and biases over frozen embeddings. A principled, group-wise decomposition of the Brier score highlights within-group variance as a major driver of residual miscalibration; incorporating a variance penalty contracts intra-group spreads and improves proper losses in practice. To decouple training from evaluation, we introduced *Logit-Cluster Calibration Error* (LCCE), a fixed-partition metric in logit space that estimates the reliability term without a trainable grouping head. Across large-scale logs and the AdAuction dataset with oracle CTR, VR-SAG consistently reduces calibration error relative to strong baselines while preserving production constraints on latency and memory.

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

# A  DECOMPOSING THE BRIER SCORE: FROM MURPHY'S TO SEMANTIC GROUPING

Modern click-through-rate (CTR) systems emit dense, high-dimensional representations and near-continuous probability scores, yet their calibration is still assessed with tools that date back to the 1970s. To understand where miscalibration originates—and how targeted post-hoc corrections like SAG or VR-SAG can fix it—we dissect the Brier score into interpretable components. We proceed in two stages. First, we revisit Murphy's classical *Uncertainty–Reliability–Resolution* (UNC–REL–RES) decomposition for hard probability bins, clarifying its assumptions and statistical meaning. Second, we generalize the same algebra to *soft, semantics-aware regions* induced by a neural grouping head, which yields an additional variance–covariance term and exposes new levers for calibration improvement. Unless otherwise stated, expectations refer to *population* quantities; empirical ("sample") analogues follow by replacing expectations with averages over data.

## A.1  CLASSICAL MURPHY UNC–REL–RES DECOMPOSITION

**Setup.** Consider a binary event $Y \in \{0, 1\}$ and a probabilistic predictor that can output only *discrete* probability levels $p_r \in [0, 1]$, $r = 1, \ldots, R$. Let $\mathcal{I}_r$ be the index set of instances that received the level $p_r$, with cardinality $n_r = |\mathcal{I}_r|$, and empirical event frequency

$$o_r \; = \; \frac{1}{n_r} \sum_{i \in \mathcal{I}_r} Y_i.$$

Denote the overall empirical base rate by $\bar{Y} = \frac{1}{n} \sum_{i=1}^{n} Y_i$ with $n = \sum_r n_r$. The (empirical) Brier score is

$$\text{BS} \; = \; \frac{1}{n} \sum_{i=1}^{n} (Y_i - p_{r(i)})^2,$$

where $r(i)$ is the level applied to instance $i$.

**Derivation.** Add and subtract $o_{r(i)}$ inside the square:

$$\text{BS} = \frac{1}{n} \sum_{i=1}^{n} \left[ (Y_i - o_{r(i)}) + (o_{r(i)} - p_{r(i)}) \right]^2$$

$$= \frac{1}{n} \sum_{i=1}^{n} (Y_i - o_{r(i)})^2 + \frac{1}{n} \sum_{i=1}^{n} (o_{r(i)} - p_{r(i)})^2 + \frac{2}{n} \sum_{i=1}^{n} (Y_i - o_{r(i)})(o_{r(i)} - p_{r(i)}). \quad (13)$$

Inside any fixed category $r$, $o_r$ is constant, hence $\sum_{i \in \mathcal{I}_r} (Y_i - o_r)(o_r - p_r) = 0$ and the cross-term in equation 13 vanishes. Grouping the remaining terms by $r$ yields (Murphy, 1973)

$$\text{BS} \; = \; \underbrace{\bar{Y}(1 - \bar{Y})}_{\text{UNC}} + \underbrace{\sum_{r=1}^{R} \frac{n_r}{n} (p_r - o_r)^2}_{\text{REL}} - \underbrace{\sum_{r=1}^{R} \frac{n_r}{n} (o_r - \bar{Y})^2}_{\text{RES}}. \quad \text{(Murphy)}$$

Here, UNC is the irreducible Bernoulli variance, REL penalizes mismatch between $p_r$ and $o_r$, and RES rewards partitions whose empirical frequencies $o_r$ deviate from the global base rate $\bar{Y}$ (hence the negative sign). Because forecasts are constant within each category, the within-category variance of the scores is zero and no extra term appears. The corresponding population identity is obtained by replacing empirical averages with expectations.

### A.2 GROUPING DECOMPOSITION WITH SOFT SEMANTIC REGIONS

**Hard vs. soft grouping.** Classical reliability diagrams assume a *hard* partition: each instance $x$ belongs to exactly one bin $G_k$ with indicator $\mathbf{1}_{G_k}(x) \in \{0,1\}$. In SAG/VR-SAG, the grouping head instead assigns a *soft* membership

$$q_k(x) \;=\; \big[\mathrm{softmax}\big(z(x)^\top W + b\big)\big]_k, \qquad 0 < q_k(x) < 1, \quad \sum_{k=1}^{K} q_k(x) = 1.$$

All "conditional" quantities below are interpreted as *soft* conditionals induced by $q_k$. For any random variable $Z = Z(x,y)$ define the (population) soft mass $w_k := \mathbb{E}[\, q_k(x)\,]$ and the soft conditional expectation

$$\mathbb{E}[Z \mid G_k] := \frac{\mathbb{E}\big[Z\, q_k(x)\big]}{\mathbb{E}\big[q_k(x)\big]} \;=\; \frac{\mathbb{E}\big[Z\, q_k(x)\big]}{w_k}.$$

With this convention,

$$\pi_k := \mathbb{E}[Y \mid G_k] = \frac{\mathbb{E}\,[Y\, q_k(x)]}{w_k}, \qquad \mu_k := \mathbb{E}[\hat{p} \mid G_k] = \frac{\mathbb{E}\,[\hat{p}\, q_k(x)]}{w_k}. \qquad \text{(soft-stats)}$$

The within–group variance and covariance are defined analogously:

$$\sigma_k^2 := \mathrm{Var}(\hat{p} \mid G_k) = \frac{\mathbb{E}\big[(\hat{p}-\mu_k)^2 q_k(x)\big]}{w_k}, \qquad \gamma_k := \mathrm{Cov}(\hat{p}, Y \mid G_k) = \frac{\mathbb{E}\big[(\hat{p}-\mu_k)(Y-\pi_k)\, q_k(x)\big]}{w_k}.$$

Setting $q_k(x) = \mathbf{1}_{G_k}(x)$ recovers the standard hard-bin formulas.

**Step 1: Law of total expectation.** Let $S(Y,\hat{p}) = (Y - \hat{p})^2$ denote the per-instance Brier score and $\bar{\pi} := \mathbb{E}[Y]$ the global click-through rate. Then

$$\mathbb{E}[S] \;=\; \sum_k w_k\, \mathbb{E}\big[(Y-\hat{p})^2 \mid G_k\big]. \tag{1}$$

**Step 2: Expand the conditional score.** Since $Y^2 = Y$ for Bernoulli labels,

$$\mathbb{E}\big[(Y-\hat{p})^2 \mid G_k\big] = \pi_k - 2\big(\mu_k \pi_k + \gamma_k\big) + \big(\mu_k^2 + \sigma_k^2\big)$$
$$= \pi_k(1 - \pi_k) + (\pi_k - \mu_k)^2 + \sigma_k^2 - 2\gamma_k. \tag{2}$$

**Step 3: Aggregate and isolate UNC, REL, RES, $\Delta$.** A direct calculation shows $\sum_k w_k \pi_k (1 - \pi_k) = \bar{\pi}(1-\bar{\pi}) - \mathrm{Var}(\pi_k)$. Substituting into the result of Step 2 gives the **grouping decomposition**

$$\mathbb{E}[S] \;=\; \underbrace{\bar{\pi}\big(1 - \bar{\pi}\big)}_{\text{UNC}} + \underbrace{\sum_k w_k(\mu_k - \pi_k)^2}_{\text{REL}} - \underbrace{\mathrm{Var}(\pi_k)}_{\text{RES}} + \underbrace{\sum_k w_k(\sigma_k^2 - 2\gamma_k)}_{\Delta}.$$

Here, UNC is the irreducible Bernoulli variance at the global level, REL measures calibration error inside semantic regions, RES rewards partitions whose prevalences $\pi_k$ are far apart, and $\Delta$ captures the additional variance–covariance contribution introduced by allowing predictions to vary within each region. When every region collapses to a single forecast ($\sigma_k^2 = \gamma_k = 0$), $\Delta$ vanishes and the formula reduces to Murphy's classical UNC + REL − RES decomposition.

**Relation to the classical Murphy decomposition.** Both Murphy's UNC–REL–RES identity and the grouping decomposition above express the same Brier score as an *uncertainty* term minus a *resolution* bonus plus a *reliability* penalty; the algebraic cores coincide. The difference lies in how the data are partitioned. Murphy's derivation assumes a *hard*, forecast-value partition ($\hat{p}$ is constant inside each cell), so the within–group variance $\sigma_k^2$ and covariance $\gamma_k$ vanish, yielding only three terms. In contrast, our decomposition keeps *soft* semantic regions learned by SAG/VR-SAG, preserving a spread of predictions within each region and introducing the additional

$$\Delta = \sum_k w_k\big(\sigma_k^2 - 2\gamma_k\big),$$

which is *not* sign-definite in general and quantifies the contribution of intra-group dispersion and label–score covariance.

**Sign of $\Delta$ and when it is positive.** Recall $\Delta = \sum_k w_k\big(\sigma_k^2 - 2\gamma_k\big)$ with $\sigma_k^2 = \mathrm{Var}(\hat{p} \mid G_k) \geq 0$ and $\gamma_k = \mathrm{Cov}(\hat{p}, Y \mid G_k)$. In general, $\Delta$ is not sign-definite: if the within-group covariance $\gamma_k$ is sufficiently positive, $(\sigma_k^2 - 2\gamma_k)$ can be negative. By Cauchy–Schwarz,

$$|\gamma_k| \leq \sigma_k \sqrt{\mathrm{Var}(Y \mid G_k)} = \sigma_k \sqrt{\pi_k(1 - \pi_k)},$$

hence

$$\sigma_k^2 - 2\sigma_k\sqrt{\pi_k(1 - \pi_k)} \ \leq \ \sigma_k^2 - 2\gamma_k \ \leq \ \sigma_k^2 + 2\sigma_k\sqrt{\pi_k(1 - \pi_k)}.$$

Therefore, $\Delta$ can be negative when many groups exhibit small $\sigma_k$ but large positive $\gamma_k$. Conversely, in sparse CTR regimes with $\pi_k \ll 1$ (so $\sqrt{\pi_k(1 - \pi_k)}$ is small) and nontrivial within-group spread $\sigma_k^2 > 0$, the lower bound is often close to $\sigma_k^2$, making $\Delta$ frequently positive in practice.

**Advantages of the grouping view.**

- **Model-aware slicing.** Regions are induced from the backbone embedding, aligning with latent user–ad semantics rather than arbitrary probability intervals.

- **Variance-aware diagnostics.** The extra $\Delta$ term reveals when score spread (large $\sigma_k^2$) or score–label coupling (large $|\gamma_k|$) dominates the Brier loss, motivating variance-reduction strategies such as VR-SAG.

- **Practicality.** A lightweight grouping head and $K(m+2)$ scalars suffice at inference, meeting strict latency and memory budgets while improving REL and overall Brier score in offline and simulated evaluations.

## B  GROUPING DECOMPOSITION FOR THE NEGATIVE LOG–LIKELIHOOD

We now derive an exact grouping decomposition for the *negative log–likelihood* (cross-entropy)

$$S_{\mathrm{NLL}}(Y, \hat{p}) \ = \ -\big[Y \log \hat{p} + (1 - Y) \log(1 - \hat{p})\big],$$

using the same soft semantic regions $G_k$ and per–region statistics $\pi_k = \mathrm{Pr}(Y{=}1 \mid G_k)$, $\mu_k = \mathbb{E}[\hat{p} \mid G_k]$, $\sigma_k^2 = \mathrm{Var}(\hat{p} \mid G_k)$, and $\gamma_k = \mathrm{Cov}(\hat{p}, Y \mid G_k)$ introduced earlier.

**Step 1: Law of total expectation.**

$$\mathbb{E}[S_{\mathrm{NLL}}] \ = \ \sum_k w_k \, \mathbb{E}\big[S_{\mathrm{NLL}} \mid G_k\big], \qquad w_k = \mathrm{Pr}(G_k). \tag{14}$$

**Step 2: Exact per–group expansion with integral remainder.** Fix a group $G_k$ and write $\delta := \hat{p} - \mu_k$. Consider the convex function $\phi_Y(p) = -Y \log p - (1 - Y) \log(1 - p)$ on $p \in (0, 1)$. By Taylor's theorem with the *integral form of the remainder*, for any $p = \mu_k + \delta$,

$$\phi_Y(p) \ = \ \phi_Y(\mu_k) + \phi_Y'(\mu_k)\,\delta + \int_0^1 (1 - t)\,\phi_Y''\big(\mu_k + t\delta\big)\,\delta^2\,dt,$$

where $\phi'_Y(\mu) = -\frac{Y}{\mu} + \frac{1-Y}{1-\mu}$ and $\phi''_Y(\xi) = \frac{Y}{\xi^2} + \frac{1-Y}{(1-\xi)^2} \geq 0$. Taking the conditional expectation given $G_k$ yields

$$\mathbb{E}\big[S_{\text{NLL}} \mid G_k\big] = \underbrace{\mathbb{E}\big[\phi_Y(\mu_k) \mid G_k\big]}_{\text{constant at } \mu_k} + \underbrace{\mathbb{E}\big[\phi'_Y(\mu_k)\,\delta \mid G_k\big]}_{\text{linear (covariance) term}} + \underbrace{\mathbb{E}\bigg[\int_0^1 (1-t)\,\phi''_Y\big(\mu_k + t\delta\big)\,\delta^2\,dt \; \bigg| \; G_k\bigg]}_{=:\, R_k \,\geq\, 0}.$$

The constant term simplifies to the cross-entropy between Bernoulli$(\pi_k)$ and Bernoulli$(\mu_k)$,

$$\mathbb{E}\big[\phi_Y(\mu_k) \mid G_k\big] = -\pi_k \log \mu_k - (1 - \pi_k)\log(1 - \mu_k) = H(\pi_k) + \text{KL}\big(\pi_k \| \mu_k\big),$$

where $H(p) = -p \log p - (1-p)\log(1-p)$. For the linear term, using $\mathbb{E}[\delta \mid G_k] = 0$ and $\mathbb{E}[Y\delta \mid G_k] = \gamma_k$ gives

$$\mathbb{E}\big[\phi'_Y(\mu_k)\,\delta \mid G_k\big] = -\frac{\gamma_k}{\mu_k} - \frac{1}{1 - \mu_k}\mathbb{E}\big[(1 - Y)\delta \mid G_k\big] = -\frac{\gamma_k}{\mu_k} + \frac{\gamma_k}{1 - \mu_k} = -\frac{\gamma_k}{\mu_k(1 - \mu_k)}.$$

Hence, for each group,

$$\mathbb{E}\big[S_{\text{NLL}} \mid G_k\big] = H(\pi_k) + \text{KL}\big(\pi_k \| \mu_k\big) - \frac{\gamma_k}{\mu_k(1 - \mu_k)} + R_k, \qquad R_k \geq 0. \qquad (15)$$

**Step 3: Isolate UNC, REL, RES, and the heterogeneity block.** Let $\bar\pi = \sum_k w_k \pi_k = \mathbb{E}[Y]$ be the global prevalence. Using the entropy identity $H(\bar\pi) - \sum_k w_k H(\pi_k) \geq 0$, combine equation 14 and equation 15 to obtain

$$\mathbb{E}[S_{\text{NLL}}] = \underbrace{H(\bar\pi)}_{\text{UNC}} + \underbrace{\sum_k w_k\,\text{KL}\big(\pi_k \| \mu_k\big)}_{\text{REL}} - \underbrace{\Big(H(\bar\pi) - \sum_k w_k H(\pi_k)\Big)}_{\text{RES}}$$

$$+ \underbrace{\sum_k w_k\,R_k}_{\text{curvature–weighted variance } (\geq 0)} - \underbrace{\sum_k w_k\,\frac{\gamma_k}{\mu_k(1 - \mu_k)}}_{\text{covariance correction}}.$$

This is an *exact* decomposition: the first three blocks match the classical UNC + REL − RES structure, while the last two terms together play the role of a heterogeneity component. When scores are constant within each region ($\sigma_k^2 = \gamma_k = 0$), we have $R_k = 0$ and the covariance term vanishes, recovering the Murphy-style three-term form. The curvature factor $1/\{\mu_k(1-\mu_k)\}$ shows that the NLL is more sensitive to within-group dispersion and score–label coupling when $\mu_k$ is near 0 or 1.

# C  THE STATISTICAL COMPARISON BETWEEN CALIBRATION METRICS

This section provides a comprehensive statistical analysis of calibration metrics, supplementing the main text's findings with detailed evaluations of metric consistency

These results validate the theoretical insights presented in the main text and offer practical guidance for metric selection in real-world calibration tasks.

## C.1  RANK CORRELATION AND VARIANCE CONSISTENCY

To assess the consistency of calibration metrics, we extended the Spearman correlation analysis from the main text to include additional metrics and conducted 5000 Monte Carlo simulations across varying data distributions. Additional metrics are listed below:

**Sufficient-information metrics.** These metrics quantify the direct discrepancy between predicted probabilities $\hat{p}$ and true click rates $\pi$, and only when ground-truth CTR is available.

- **Brier$^+$ score**:

$$\text{Brier}^+ = \frac{1}{N}\sum_{i=1}^N |\pi_i - \hat{p}_i|$$

- **KL divergence**:

$$\text{KL}(\pi\|\hat{p}) = \frac{1}{N}\sum_{i=1}^{N}\pi_i \log\left(\frac{\pi_i}{\hat{p}_i}\right)$$

- **Generalized KL divergence**:

$$\text{KL}_\alpha(\pi\|\hat{p}) = \frac{1}{N}\sum_{i=1}^{N}\frac{1}{\alpha(\alpha-1)}\left(\pi_i^\alpha + (\alpha-1)\pi_i - \alpha\pi_i\hat{p}_i^{1-\alpha}\right)$$

- **Chebyshev distance**:

$$\max_i |\pi_i - \hat{p}_i|$$

- **Pearson correlation**:

$$\rho(\pi,\hat{p}) = \frac{\sum_{i=1}^{N}(\pi_i - \bar{\pi})(\hat{p}_i - \bar{\hat{p}})}{\sqrt{\sum_{i=1}^{N}(\pi_i - \bar{\pi})^2 \sum_{i=1}^{N}(\hat{p}_i - \bar{\hat{p}})^2}}$$

- **Spearman correlation**:

$$\rho_s(\pi,\hat{p}) = \frac{\sum_{i=1}^{N}(r_i - \bar{r})(q_i - \bar{q})}{\sqrt{\sum_{i=1}^{N}(r_i - \bar{r})^2 \sum_{i=1}^{N}(q_i - \bar{q})^2}}$$

**Insufficient-information metrics** These metrics quantify the direct discrepancy between predicted probabilities $\hat{p}$ and labels $y_i$, and only when ground-truth CTR is unavailable.

- **MSE**:

$$\text{MSE} = \frac{1}{N}\sum_{i=1}^{N}(y_i - \hat{p}_i)^2$$

- **MAE**:

$$\text{MAE} = \frac{1}{N}\sum_{i=1}^{N}|y_i - \hat{p}_i|$$

- **ECE$^+$**: $G_k$ denotes equal-frequency bins

$$\text{ECE}^+ = \sum_{k=1}^{K}\frac{|G_k|}{N}\left|\frac{1}{|G_k|}\sum_{i\in G_k}\hat{p}_i - \frac{1}{|G_k|}\sum_{i\in G_k}y_i\right|$$

- **LCCE rand group**:

$$\text{LCCE}_{\text{rand}} = \sum_{k=1}^{K}w_k|\mu_k - \pi_k|, \quad w_k = \frac{|G_k|}{N}$$

- **LCCE field group**:

$$\text{LCCE}_{\text{field}} = \sum_{k=1}^{K}w_k|\mu_k - \pi_k|$$

- **Bias and absolute Bias**:

$$\text{bias} = \frac{\bar{\hat{p}}}{\bar{y}}, \quad \text{abs\_bias} = \sum_{k=1}^{K}w_k\left|\frac{\bar{\hat{p}}_k}{\bar{y}_k}\right|$$

Table 3,4 and 5 presents the rank correlation and variance consistency of metrics.

The three figures collectively illustrate the consistency and reliability of calibration metrics across different statistical properties. The first heatmap 3 shows that group-based metrics like LCCE exhibit stronger Spearman correlation (up to 0.94) with the true Brier score compared to point-wise

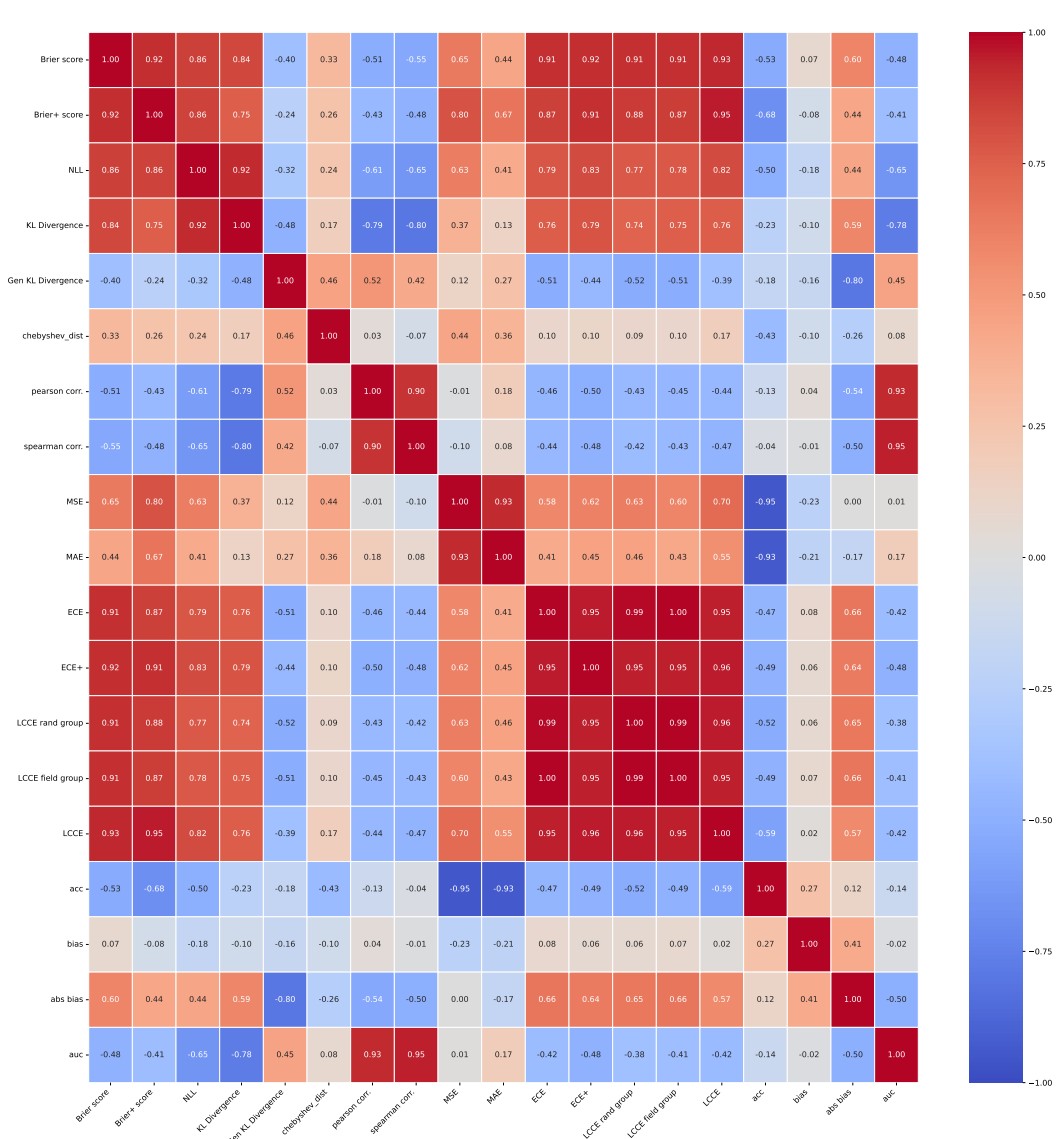

Figure 3: Spearman correlation heatmap of calibration metrics. Warmer colors indicate higher correlation, with group-based metrics LCCE showing stronger alignment with true Brier score.

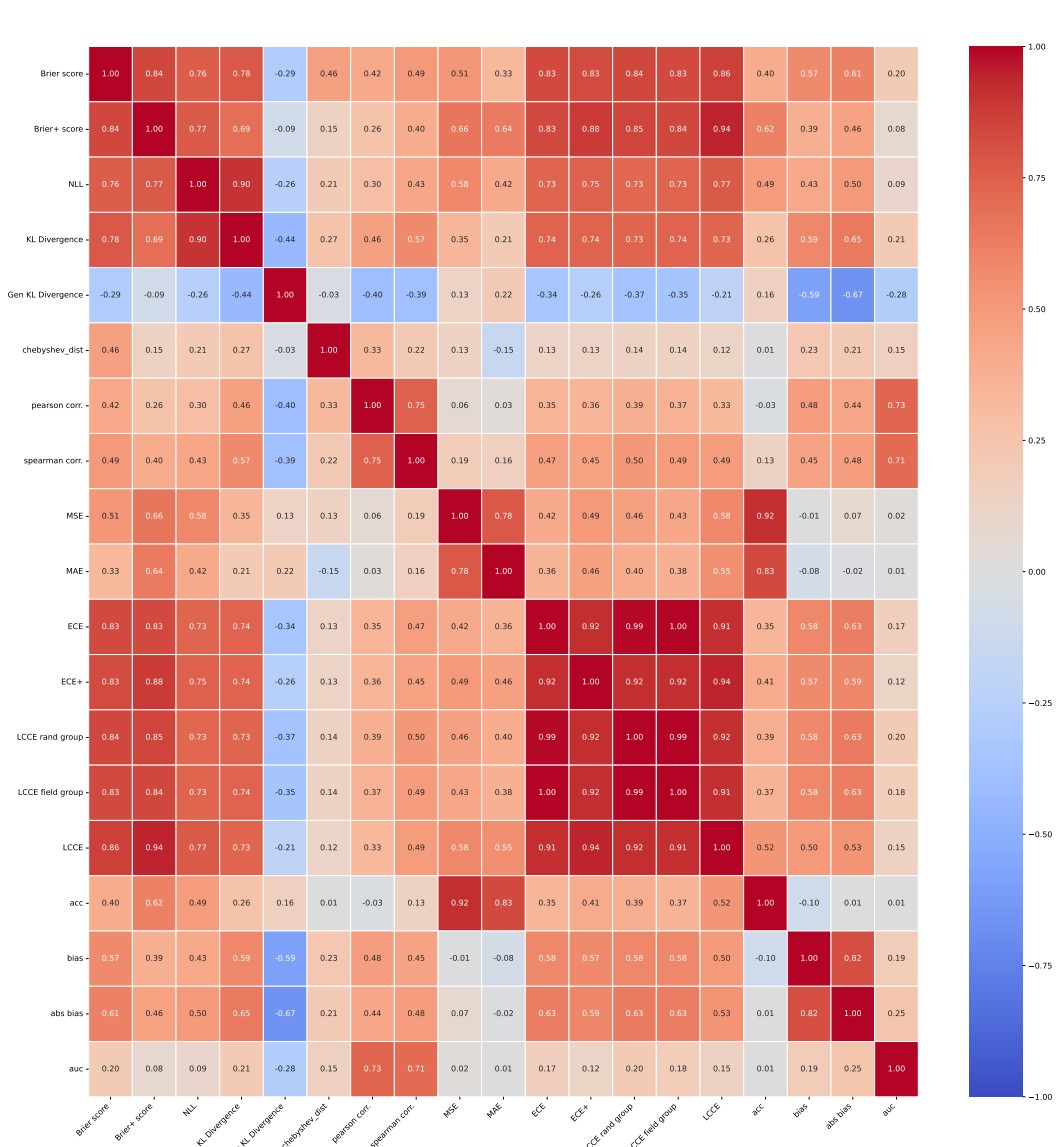

Figure 4: Spearman correlation heatmap of the standard deviations (std) of calibration metrics. If a metric is consistent with the Brier score, its moment functions (including the standard deviation) should also align. LCCE demonstrates stronger consistency (0.86 and 0.94) than other metrics in capturing the second-order statistical properties of calibration error, as evidenced by its higher correlation with the Brier score's standard deviation.

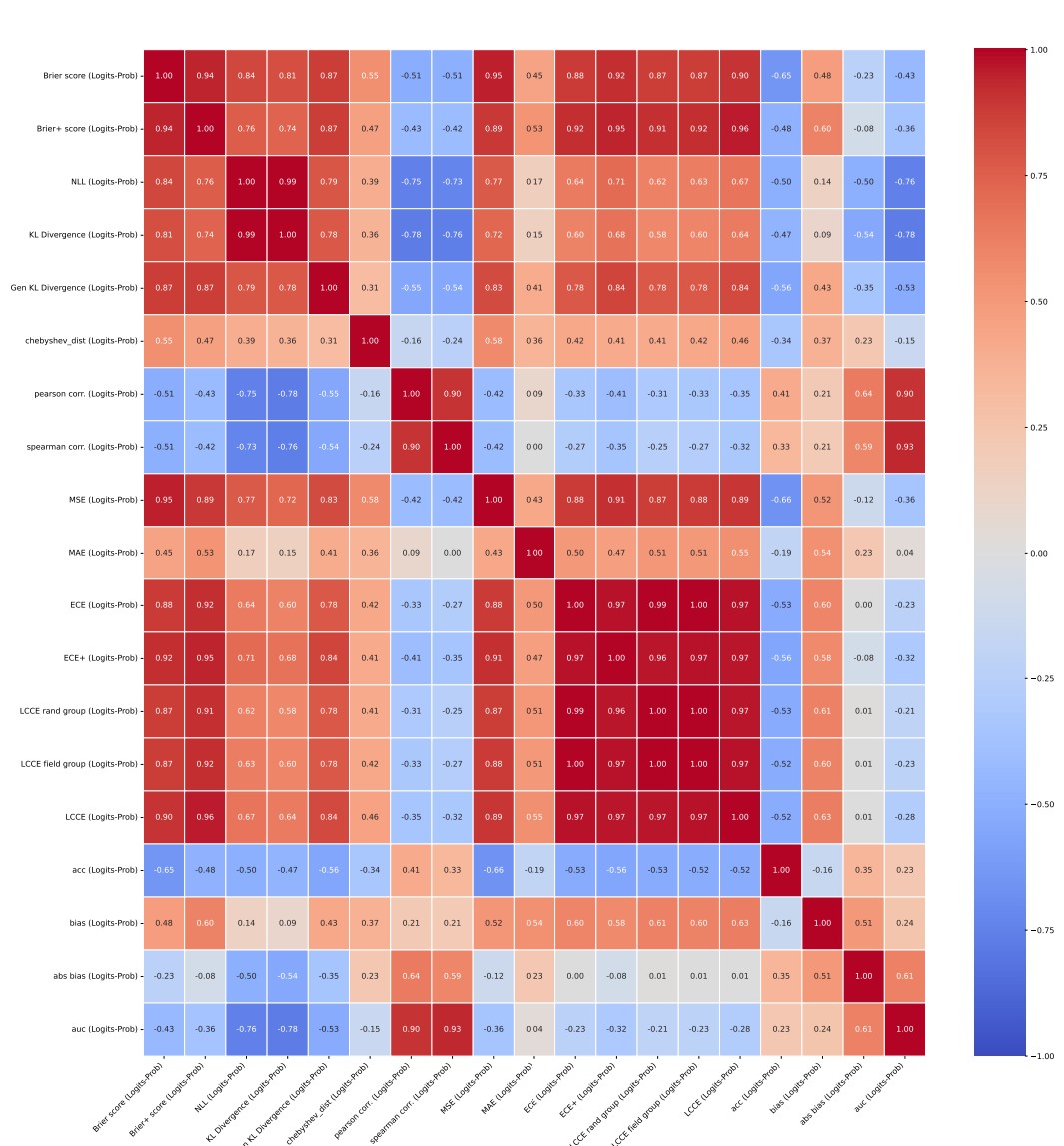

Figure 5: Spearman correlation heatmap of metric improvements (calibrated - uncalibrated). LCCE exhibits the highest consistency with Brier score improvements, followed by equal-frequency ECE. The semantic grouping of LCCE outperforms manual binning, as it learns latent structures to capture true calibration errors, whereas equal-frequency binning lacks semantic interpretability and may miss fine-grained discrepancies.

metrics, highlighting their superiority in capturing calibration errors. The second figure, focusing on standard deviations, reveals that LCCE maintains higher consistency (0.86) with the Brier score's second-order statistics, demonstrating its stability in quantifying calibration uncertainty. The third heatmap, analyzing metric improvements (calibrated - uncalibrated), further confirms LCCE's dominance 0.92), outperforming equal-frequency ECE (0.85) by leveraging semantic grouping to learn latent structures. This allows LCCE to capture fine-grained discrepancies missed by manual binning, underscoring the importance of data-driven grouping in enhancing calibration assessment accuracy.

## C.2 CONVERGENCE PROPERTIES

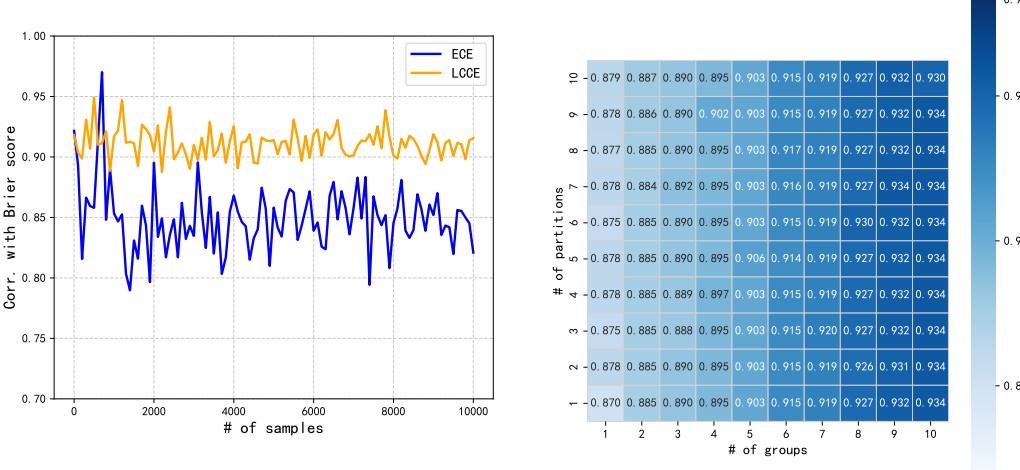

(a) Spearman correlation between ECE/LCCE and Brier score across varying sample sizes.

(b) Spearman correlation of LCCE under different hyperparameter configurations.

Figure 6: Convergence behavior of calibration metrics with sample size (a) and LCCE stability under hyperparameter variations (b)

Figure 6(a) demonstrates the convergence of ECE and LCCE to stable values as the sample size increases, with LCCE exhibiting higher Spearman correlation with the Brier score than ECE—consistent with prior findings. The lower variance of LCCE arises from its default configuration of 4 partitions, effectively averaging results from 4 Monte Carlo samplings to reduce estimation noise.

## C.3 HYPERPARAMETER SENSITIVITY

In Figure 6(b), the number of partitions shows minimal impact on LCCE performance, while increasing the number of groups systematically improves metric accuracy. This stability stems from LCCE's k-means clustering with centroid compatibility: when the number of groups is large, proximally similar clusters are automatically merged, preventing overfitting to noise. This feature makes LCCE robust to group number selection, enabling reliable calibration assessment across diverse data scales.

## D TRAINING COMPLEXITY

Because the compared calibration methods differ in both model structure and available implementations, asymptotic complexity alone is not very informative about their practical cost. Instead, we report wall–clock time for the full calibration pipeline (training on $D_{\mathrm{val}}$ and evaluating on $D_{\mathrm{ho}}$) under a shared hardware setup. For all methods we rely on the authors' public implementations or faithful reimplementations, and run them on the same machine (NVIDIA 2060 Super GPU, Intel Core i7-9700 CPU, 8GB RAM).

Table 2: Training+testing time (minutes) for different calibration methods on each dataset.

|  | AdAuction | ALICCP | ALIEXP |
|---|---|---|---|
| Histogram binning | 15 | 39 | 45 |
| Isotonic regression | 17 | 35 | 41 |
| Platt scaling | 17 | 35 | 44 |
| Temperature scaling | 20 | 37 | 44 |
| SAG+PS | 29 | 65 | 52 |
| SAG+TS | 28 | 64 | 53 |
| VR-SAG+PS | 32 | 97 | 57 |
| VR-SAG+TS | 31 | 93 | 59 |

As shown in Table 2, classical one-dimensional post–hoc calibrators (histogram binning, isotonic regression, Platt and temperature scaling) are the fastest, typically finishing within tens of minutes even on the largest dataset. Grouping-based methods incur additional cost from learning the grouping head and per-group parameters: SAG is moderately slower, and VR-SAG adds a further overhead due to the variance penalty and the extra statistics it requires. Nevertheless, the total cost remains in the order of tens of minutes per dataset, and the incremental overhead of VR-SAG over SAG is small compared to the overall backbone training time in realistic CTR/CVR systems.

## E  OTHER METRICS AND ANALYSIS

In industrial online applications, the accurate assessment of model prediction discrepancies, including overestimations and underestimations, is of paramount importance. Among the various evaluation metrics, the predict click rate over click rate (pcoc) metric and the bias metric are frequently employed to gauge the calibration quality of model predictions. The bias metric, defined as

$$bias(\hat{p}, y) = \frac{\bar{\hat{p}}}{\bar{y}}, \ abs\_bias(\hat{p}, y) = \sum_k w_k |\frac{\bar{\hat{p}_k}}{\bar{y_k}}|$$

where $k$ denotes grouping by media id (i.e., field-wise statistics). Bias directly quantifies the relative deviation between predicted and actual values, offering a concise and intuitive measure for identifying systematic overestimation or underestimations in model outputs. This metric captures the overall directional trend in expectations, though it incurs information loss—particularly due to the cancellation problem. For instance, positive and negative biases across different media groups may offset each other, masking true calibration errors.

The absolute bias metric mitigates this cancellation issue by summing weighted absolute deviations. Consider a scenario where one media group exhibits overestimation and another underestimation:their respective biases might cancel out, yielding a deceptively low overall bias. In contrast, the absolute bias captures such discrepancies by emphasizing the magnitude of deviations, ensuring that mis-calibrations in opposite directions are not overlooked. Specifically, a positive bias value indicates model overestimation, while a negative value signals underestimation, with the absolute bias providing a more robust measure of calibration accuracy across grouped fields.

In the following analysis, we incorporate these key industrial metrics to comprehensively evaluate the performance of each comparative method.

Table 3 presents the bias and absolute bias metrics of various calibration methods across three industrial datasets. For the AdAuction dataset, the VR-SAG+TS method achieves the smallest bias (0.0055), demonstrating its superiority in mitigating overall prediction deviation. In terms of absolute bias, which avoids cancellation of positive and negative errors, the values are generally higher than the bias metrics, highlighting the importance of using absolute bias to capture true calibration errors. Specifically, VR-SAG+PS obtains the minimum absolute bias (1.0742), outperforming other methods in quantifying the magnitude of prediction discrepancies without direction offset.For the ALICCP and AE datasets, the bias values of most calibration methods are controlled within the order of $10^{-3}$, suggesting negligible overall deviation at first glance. However, this apparent "smallness" of bias metrics is misleading due to potential cancellation effects across different media groups.

Table 3: Comparison of calibration methods. We utilized bold font to highlight the statistically superior ($p < 0.05$) results.

| | AdAuction | | ALICCP | | AE | |
|---|---|---|---|---|---|---|
| Method | bias | abs_bias | bias | abs_bias | bias | abs_bias |
| Uncal | 1.5866 | 13.2670 | -0.039843 | 0.0398 | -0.964428 | 0.9644 |
| Histogram binning | 0.0061 | 4.5772 | -0.001060 | 0.0042 | 0.002417 | 0.0046 |
| Isotonic regression | 0.0086 | 4.1161 | -0.001093 | 0.0042 | 0.002474 | 0.0048 |
| Platt scaling | 0.0074 | 4.2505 | -0.001073 | 0.0043 | 0.002780 | 0.0048 |
| Temperature scaling | 0.0079 | 4.2504 | -0.001071 | 0.0042 | -0.018733 | 0.0187 |
| SAG+PS | 0.0075 | 1.7436 | -0.000213 | 0.0036 | 0.000244 | 0.0034 |
| SAG+TS | 0.0088 | 1.8910 | -0.000202 | 0.0035 | -0.001183 | 0.0061 |
| VR-SAG+PS | 0.0074 | **1.0742** | -0.000213 | 0.0036 | 0.000193 | **0.0027** |
| VR-SAG+TS | **0.0055** | 1.7331 | -0.000213 | 0.0035 | **-0.000104** | 0.0054 |

The absolute bias metrics, though also low in magnitude, provide a more reliable assessment by emphasizing the cumulative deviation. For example, in the AE dataset, VR-SAG+PS achieves the smallest absolute bias (0.0027), indicating its robustness in handling grouped calibrations without masking errors through direction cancellation.These results underscore the critical role of absolute bias in industrial calibration evaluations, as it overcomes the limitation of traditional bias metrics that may obscure true calibration errors due to positive-negative cancellation. While the bias values on ALICCP and AE datasets suggest satisfactory performance, the absolute bias metrics reveal the nuanced differences in calibration quality, guiding the selection of more robust methods for practical applications.

### E.1 ACCURACY PRESERVING

Not all calibration methods guarantee the preservation of predictive accuracy, as some may risk degrading metrics such as accuracy or AUC. To assess this, we measure the proportional difference in accuracy and AUC between the calibrated outputs and the backbone model's logits. For each calibration method, we compute:

$$\text{AccDiff} = \frac{\text{Acc(calibrated)} - \text{Acc(logits)}}{\text{Acc(logits)}},$$

$$\text{AUCDiff} = \frac{\text{AUC(calibrated)} - \text{AUC(logits)}}{\text{AUC(logits)}},$$

where Acc($\cdot$) denotes classification accuracy and AUC($\cdot$) denotes the area under the receiver operating characteristic curve. This metric quantifies the relative change in predictive performance induced by calibration, enabling a systematic evaluation of accuracy degradation risks. Positive values indicate performance improvement, while negative values signal potential accuracy loss—highlighting the trade-off between calibration quality and predictive fidelity.

Table 4 illustrates the relative changes in accuracy and AUC between calibrated outputs and backbone logits. Notably, Platt Scaling (PS) and Temperature Scaling (TS) demonstrate strict accuracy preservation, as their AccDiff and AUCDiff values for most datasets approach zero, indicating minimal disruption to the original prediction ordering. This consistency aligns with their parametric calibration nature, which adjusts confidence scores without altering the rank of predictions.

Histogram Binning (HB) exhibits the most pronounced AUC degradation, particularly on the AE dataset (-0.2857), attributed to its mechanism of assigning uniform predictions within each bin. This process forfeits fine-grained relative ordering, as all samples in a bin are forced to share the same estimated value, thereby compromising the discriminative power essential for AUC.

For AdAuction and ALICCP, calibration methods generally yield marginal improvements in both accuracy and AUC. The subtle boosts on ALICCP (e.g., up to 0.0015 in AccDiff) are noteworthy given its inherently lower baseline AUC, suggesting that calibration effectively refines prediction

Table 4: Accuracy preserving measure of calibration methods.

| | AdAuction | | ALICCP | | AE | |
|---|---|---|---|---|---|---|
| Method | AccDiff | AUCDiff | AccDiff | AUCDiff | AccDiff | AUCDiff |
| Uncal | 0.0000 | 0.0000 | 0.0000 | 0.0000 | 0.0000 | 0.0000 |
| Histogram binning | 0.0007 | -0.1783 | 0.0015 | -0.0849 | 0.0000 | -0.2857 |
| Isotonic regression | 0.0007 | -0.0167 | 0.0015 | -0.0002 | 0.0000 | 0.0006 |
| Platt scaling | 0.0004 | 0.0000 | 0.0015 | 0.0000 | -0.0001 | 0.0000 |
| Temperature scaling | 0.0004 | 0.0000 | 0.0015 | 0.0000 | -0.0001 | 0.0000 |
| SAG+PS | 0.0005 | 0.0008 | 0.0015 | -0.0533 | -0.0001 | -0.0327 |
| SAG+TS | 0.0006 | 0.0009 | 0.0015 | 0.0035 | -0.0001 | -0.0404 |
| VR-SAG+PS | 0.0007 | 0.0037 | 0.0015 | -0.0643 | -0.0001 | -0.0153 |
| VR-SAG+TS | 0.0007 | 0.0052 | 0.0015 | 0.0017 | -0.0001 | -0.0149 |

rankings even in scenarios with modest initial performance. In contrast, the AE dataset shows slight accuracy degradation (e.g., -0.0001 for TS), likely due to overfitting during calibration on its specific data distribution.

Overall, all methods induce minimal changes in predictive correctness, with most AccDiff and AUCDiff values bounded within ±0.005. This stability highlights the balance between calibration quality and accuracy preservation, confirming that the proposed methods maintain the backbone model's predictive fidelity while enhancing probability calibration.

## F    ON THE POSSIBILITY OF EMPTY GROUPS AND OUR HANDLING

Because groups are learned from data, it is theoretically possible that a group receives no training samples. In all our main experiments—where the number of groups $K$ is set to a few *dozens*—we did *not* observe any empty groups. Hence, under practical choices of $K$, this issue has negligible impact.

To probe the worst case, we further ran stress tests with substantially larger $K$. Even in this extreme setting, the effect on overall calibration metrics (ECE/LCCE and, where applicable, Brier variants) was very small, and the trends reported in the main paper remained unchanged.

Our implementation adopts a safe default: per-group calibrators are initialized to the identity and remain unchanged if a group lacks training support. Concretely, for group $k$ we set

$$\tau_k = 1, \qquad \beta_k = 0,$$

so the calibrated probability reduces to the backbone score $\tilde{p} = \sigma(o/\tau_k + \beta_k) = \sigma(o)$. This *no-change* fallback prevents unintended shifts and guarantees that unsupported groups cannot degrade predictions.

If additional robustness is desired in very large-scale deployments, one may (i) merge groups below a minimum support into the nearest supported group, or (ii) back off to a global calibrator. These options require no changes to VR-SAG's core design and preserve its latency/memory profile.

## G    ADDITIONAL METRICS—NLL AND AUC

To complement the calibration metrics reported in the main text, we provide *negative log-likelihood* (NLL)—a proper scoring rule—and AUC for all main experiments. The conclusions are unchanged: VR-SAG attains the strongest performance on NLL in the majority of settings while maintaining competitive ranking quality (AUC). In particular, VR-SAG+TS achieves the lowest NLL on **Al-iCCP** and **AE**, and its NLL on **AdAuction** is on par with the best baseline (difference $\leq 0.0006$) while preserving top-tier AUC. These results mirror the improvements observed under LCCE and corroborate that variance control improves *proper* loss without harming ranking.

Table 5: Negative Log-Likelihood (NLL; lower is better) and AUC across datasets. VR-SAG consistently matches or surpasses strong baselines on NLL while keeping AUC competitive.

| Method | AdAuction | | AliCCP | | AE | |
|---|---|---|---|---|---|---|
| | NLL | AUC | NLL | AUC | NLL | AUC |
| Uncal | 0.3444 | 0.8212 | 0.1692 | 0.6130 | 0.3853 | 0.6302 |
| Histogram binning | 0.3207 | 0.6747 | 0.1702 | 0.6101 | 0.3933 | 0.6079 |
| Isotonic regression | 0.3199 | 0.8074 | 0.1696 | 0.6128 | 0.3922 | 0.6085 |
| Platt scaling | 0.3187 | 0.8212 | 0.1691 | 0.6130 | 0.3535 | 0.6302 |
| Temperature scaling | 0.3187 | 0.8212 | 0.1691 | 0.6130 | 0.3519 | 0.6303 |
| SAG+PS | **0.3183** | 0.8146 | 0.1658 | 0.6618 | 0.3588 | 0.6568 |
| SAG+TS | 0.3185 | 0.8154 | 0.1640 | 0.6896 | 0.3459 | 0.6502 |
| VR-SAG+PS | 0.3191 | 0.8210 | 0.1625 | **0.6897** | 0.3372 | 0.6652 |
| VR-SAG+TS | 0.3193 | **0.8221** | **0.1621** | 0.6885 | **0.3297** | **0.6660** |

**Summary.** NLL results align with LCCE improvements: variance-reduced semantic grouping yields better-calibrated probabilities under a proper loss, and AUC remains competitive—confirming that VR-SAG sharpens probability quality without sacrificing ranking.

## H PRACTICAL NOTES ON LCCE AND THE SIMULATOR

### H.1 LCCE AT SCALE AND UNDER DISTRIBUTION SHIFT

LCCE applies $K$-means to one-dimensional logits, which makes clustering practical even with very large impression volumes. In our experience it is unnecessary to cluster on all impressions: stable centroids can be estimated from a modest random subsample, after which computing LCCE reduces to assigning each impression to its nearest centroid and aggregating the resulting statistics. The overall cost therefore consists of a small-sample $K$-means run to obtain $K$ centroids, followed by an $O(N \times K)$ pass for assignment and aggregation over $N$ impressions. This workflow is typically fast enough per evaluation window, so incremental or streaming variants are not required in practice, although standard accelerations such as mini-batch $K$-means, simple one-dimensional initializations, and vectorized or parallel distance computations can further reduce wall-clock time without changing the metric. To quantify sampling effects, we evaluate LCCE as a function of the clustering subsample size and observe rapid stabilization once the subsample reaches $10^5$; even $10^3$–$10^4$ samples already provide a close approximation, as shown in Table 6. To handle non-stationarity, we simply re-estimate the one-dimensional centroids whenever LCCE is computed. This keeps pace with distributional drift at modest cost—comparable to common binning-based calibration metrics—and works well on rolling evaluation windows, where centroids change little under slow drift and adapt immediately when drift accelerates.

Table 6: Influence of clustering subsample size on LCCE (lower is better).

| Subsample size | LCCE |
|---|---|
| $10^2$ | 0.012309877 |
| $10^3$ | 0.021076037 |
| $10^4$ | 0.020131670 |
| $10^5$ | 0.019488912 |
| $10^6$ | 0.019336675 |
| $10^7$ | 0.019336664 |

### H.2 SIMULATOR GROUND-TRUTH CTRS: RATIONALE AND VALIDATION

Real-world logs do not expose true click probabilities, so a controlled environment is needed to compare calibration methods and metrics on equal footing. Our simulator serves this purpose as an explicit, model-based proxy: it is trained on production data and validated to match production feature distributions and CTR distributions both globally and across salient feature groups. These

statistical checks support that the generated labels are of sufficient quality for comparative calibration studies, while acknowledging that latent user CTRs remain unobservable in principle. To encourage transparency and reuse, we release the resulting AdAuction dataset and accompanying calibration code so that our results can be reproduced and future methods can be compared fairly, while keeping the internal simulator itself proprietary.

### H.3 VERIFYING THE EFFECT OF THE INTRA-GROUP VARIANCE CONSTRAINT

As evidenced in Table 1, VR-SAG improves overall calibration by minimizing within-group variance. A visual grouping analysis further shows that clusters learned by VR-SAG exhibit markedly more homogeneous score distributions within each group than those learned without the variance-reduction term.

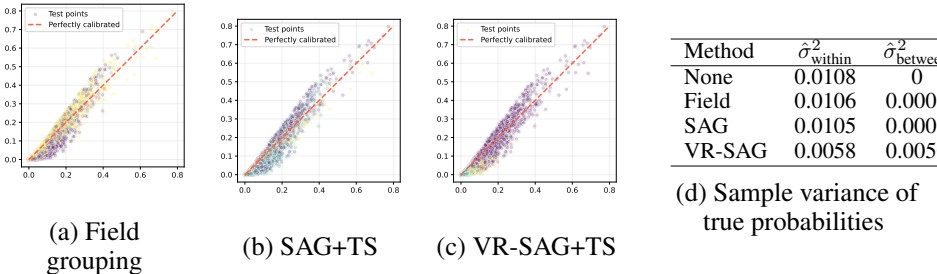

(a) Field grouping      (b) SAG+TS      (c) VR-SAG+TS

| Method | $\hat{\sigma}^2_{\text{within}}$ | $\hat{\sigma}^2_{\text{between}}$ |
|--------|------|------|
| None   | 0.0108 | 0 |
| Field  | 0.0106 | 0.0001 |
| SAG    | 0.0105 | 0.0002 |
| VR-SAG | 0.0058 | 0.0050 |

(d) Sample variance of true probabilities

Figure 7: Grouping visualization. Backbone estimates (mean = 0.089) vs. true probabilities (mean = 0.073), with points color-coded by class labels. A 1% uniform downsampling is used for clarity. VR-SAG yields the lowest within-group variance (0.0058), indicating superior group separation.

Perfect calibration requires grouping that maximizes within-group homogeneity (minimal variance of true probabilities) and between-group separability (distinct mean true probabilities). As shown in Figure 7, VR-SAG better satisfies both criteria than rigid field grouping: its clusters are more concentrated (lower intra-group dispersion in (c)), aligning with the smallest within-group variance in panel (d). In other words, even if the groups do not correspond to single human-readable fields, they form latent regions in embedding space that are explicitly more homogeneous in true CTR and thus more meaningful for calibration.

## I REPRODUCIBILITY

To ensure the reproducibility of our paper, we have included all the necessary code for replicating the experiments in the supplementary materials. The code has been anonymized to maintain the anonymity of the review process. Instructions for running the code and specific implementation details for each method can be found in the README.md file and commented within the code itself.

### I.1 IMPLEMENTATION AND HYPER-PARAMETER TUNING.

The implementations of the comparative methods in the paper have been modified from the corresponding open-source codes of their respective papers.

Specifically, the Temperature Scaling, Histogram Binning, Beta Calibration, and Isotonic Regression methods were modified from the open-source [3] of Guo et al.(Guo et al., 2017a), and SAG method was modified from the open-source [4] of Yang et al. (Yang et al., 2023).

For some hyperparameters in the comparative methods, we follow the same setting of Yang et al. (Yang et al., 2023).

---

[3] `https://github.com/markus93/NN_calibration/blob/master/scripts/calibration/cal_methods.py`

[4] `https://github.com/ThyrixYang/group_calibration`

Table 7: Statistical overview of datasets and backbone approach.

|  | Dataset example | | | | Backbone performance | |
|---|---|---|---|---|---|---|
|  | Impressions | Clicks | CTR | Avg. true CTR | Avg. pCTR | AUC |
| Aliccp | 42M | 164K | 0.0389 | no data | 0.0373 | 0.5875 |
| Aliexpress | 22M | 574K | 0.0257 | no data | 0.0257 | 0.7681 |
| AdAuction | 15M | 451K | 0.0311 | 0.0301 | 0.0410 | 0.8903 |

To ensure rigorous evaluation, we randomly partitioned a validation set $D_{val}$ comprising 10% of the standard training data, alongside a hold-out set $D_{ho}$ consisting of 10% from the standard test set for calibration purposes. For each dataset-model combination, we executed 100 distinct test set splits to derive statistically robust results, reporting performance metrics as the average across 100 trials for each method. Paired t-tests were conducted to assess the statistical significance of observed improvements.

Hyperparameters for comparative methods were optimized following established protocols in the literature, utilizing 5-fold cross-validation. The number of groups was fixed at 3, and the number of partitions was set to 10. Adopting a tuning strategy analogous to comparative approaches, the regularization strength was parameterized as $\lambda = 0.1$ , and the group variance coefficient was specified as $\lambda_v = 0.5$

## I.2 DATASETS

we conduct a comprehensive analysis of various calibration error metrics. Then for offline experiments, our method was tested on two widely used public datasets—AliCCP(Ma et al., 2018) and AliExpress(Xu et al., 2019)—and our newly open-sourced AdAuction dataset. The Table presents the statistics of these datasets

Due to the company's open-source dataset restrictions, we have only uploaded the ALICCP and ALIEXP datasets. The ALICCP backbone logits are trained and their features are extracted from open-source code [5] and the ESMM config are used. The ALIEXP backbone logits are trained and their features are extracted from open-source code [6] and the SharedBottom config are used. Upon acceptance of the paper, we will make all the AdAuction dataset publicly available together.

## J THE USE OF LARGE LANGUAGE MODELS (LLMs)

In this article, we only use LLMs for polishing the writing and for limited searches of relevant literature.

---

[5]https://github.com/datawhalechina/torch-rechub/blob/main/examples/ranking/run_ali_ccp_multi_task.py

[6]https://github.com/datawhalechina/torch-rechub/blob/main/examples/ranking/run_aliexpress.py

