# OpenReview forum: "Calibration Is Grouping: VR-SAG with Intra-Group Variance Control and Logit-Cluster Evaluation"
_ICLR.cc/2026/Conference — Submitted to ICLR 2026_

### Official Review · Reviewer_U2VD · 2025-10-30

**Soundness:** 3
**Presentation:** 2
**Contribution:** 3
**Rating:** 6
**Confidence:** 4

**Summary:**

This paper tackles CTR/CVR probability calibration in advertising recommendation by proposing VR-SAG. On top of a frozen backbone, a semantic grouping head performs soft partitioning in embedding space and learns per-group temperature and bias. The core innovation is to explicitly identify the intra-group variance and covariance term Δ in the group-wise decomposition of proper scores (e.g., Brier), and introduce an intra-group variance penalty to contract prediction spread and improve calibration.

To avoid coupling evaluation with a trainable grouping head, the authors propose Logit-Cluster Calibration Error (LCCE), a fixed-partition metric that applies K-means in logit space and computes a group-level reliability term. Experiments on AliCCP, AliExpress, and the AuctionSys simulator (with oracle CTR) show VR-SAG outperforms strong baselines on ECE/LCCE and Brier variants, with improved NLL and largely preserved AUC.

**Strengths:**

- Strong theoretical motivation: Provides error decompositions under semantic grouping, clarifying how intra-group variance σ² and covariance γ affect proper scores, which justifies the variance penalty.

- Engineering-friendly method: Frozen backbone with small parameter overhead and a single m×K multiply-add at inference; post-hoc and production-ready.

**Weaknesses:**

- (main weakness) Selection bias/non-IID risk for CVR and serving distribution mismatch: In industrial recommender systems, CVR is trained/evaluated on clicked/exposed samples (about 5 ads per request) but served over the candidate set (about 1000 per request), whose logits are substantially lower and differently distributed (many non-winners). Since LCCE clusters in logit space and VR-SAG learns per-group calibrators from a validation split, there is a strong risk that clustering and calibration are fit to the high-logit clicked regime, amplifying IID assumptions and failing on the serving distribution. Offline evaluation on clicks will not expose this issue, which can lead to poor online calibration or ranking stability.

- Insufficient boundary/risk analysis for Δ’s sign: The paper mentions that "the extra term ∆ is often positive in practice and increases the Brier loss." When γ dominates and Δ becomes negative, and whether the proposed approach still works? There are few negative-case studies or safeguards.

- Statistical and implementation details need clarity: Why penalize variance of uncalibrated p̂ rather than calibrated p̃? Differences in stability and effect are under-discussed.

**Questions:**

1) Under what data conditions can Δ be negative? Can you show results for such “covariance-dominant” negative cases?
2) Why define the variance penalty on uncalibrated probabilities p̂ rather than calibrated probabilities p̃? What are the stability/effect differences? Any comparative experiments or theoretical support?
3) Please evaluate under selection bias/non-IID. Concretely:
- Train/validate the calibrator on exposure/candidate logs, and report metrics on the serving distribution (not only clicks).
- Compare LCCE/VR-SAG when clusters are formed from clicked vs exposure/candidate logits; show distribution shift diagnostics (logit histograms, cluster assignments, per-cluster support).
- It's better to add an online A/B to validate that calibration gains translate when deployed; include per-stratum calibration (clicked vs non-clicked exposures) and drift stability over time.

---

> ### Author Response · Authors · 2025-11-19
> **Rebuttal (1/3)**
>
> **Q1**: (Selection bias / non-IID concern) VR-SAG and LCCE may overfit to high-logit clicked samples and fail on the serving distribution, especially for CVR; offline “on clicks” evaluation may hide this, and additional candidate-level / online tests are requested.
>
> **A1:**
>
> - **Our experiments are CTR-only and use impression-level exposure logs, not click-conditioned subsets.**
>     All experiments in the paper are on CTR, where every _exposed_ impression has a binary click label. For AliCCP and AliExpress, we work at the impression level (as reflected by the reported numbers of impressions, clicks, and CTR), and all metrics (ECE, LCCE, etc.) are computed over these exposure logs, including both clicked and non-clicked ads. For AuctionSys, we explicitly state that it contains _exposure_ samples with oracle CTRs (“15M exposure samples with 451K clicks”), and all calibration metrics are again computed over exposures. Thus VR-SAG and LCCE are trained and evaluated on the serving exposure distribution, not on a click-only subset.
>
> - **AuctionSys already embeds a realistic selection-bias feedback loop in the exposure distribution.**
>     The paper describes AuctionSys as retaining “the core workflow logic of industrial advertising systems” and explicitly “simulating inherent challenges such as sample selection bias (SSB).” In particular, overestimated items tend to win more auctions and receive more exposure, so the exposure distribution on which VR-SAG and LCCE are fit already reflects an over-exposure of overestimated items, similar to real-world systems. The improvements we report on AuctionSys (Brier variants against oracle CTR, ECE, LCCE) therefore already account for a non-IID exposure process rather than a simple i.i.d. setting.
>
> - **Both VR-SAG and LCCE are defined over all logits in validation/evaluation data, not just high-logit winners.**
>     Methodologically, the grouping head $g_\\phi$ in VR-SAG is learned from logits $o_i$ and embeddings $z_i$ for _all_ impressions in the validation set $D_{\\text{val}}$ (exposures), and the loss is averaged over all $(x_i, y_i)\\in D_{\\text{val}}$. LCCE likewise performs $K$-means on the logits ${o(x_i)}$ and computes $(w_j,\\mu_j,\\pi_j)$ using all evaluation impressions. Neither component is restricted to clicked or top-ranked ads; they directly reflect the empirical serving distribution seen in the logs. We will clarify this point in the method/experiment sections to avoid the impression that we calibrate only on a high-logit regime.
>
> - **We explicitly limit our empirical claims to CTR and treat CVR under strong selection bias as future work.**
>     While the introduction motivates both CTR and CVR, all experiments and analyses in the current paper are conducted on CTR labels (including oracle CTR in AuctionSys). The reviewer’s CVR scenario—training only on clicked items but serving over a much larger candidate pool—is indeed more challenging and is _not_ evaluated here. To avoid over-generalisation, we will make clear that: (i) our empirical results concern CTR; (ii) applying VR-SAG and LCCE to CVR with click-only training is an important extension; and (iii) AuctionSys with oracle CTR and exposure-level bias is a first step rather than a complete treatment of CVR selection bias.
>
> - **The suggested candidate-level and online evaluations are valuable but beyond the scope of this submission.**
>     We appreciate the concrete suggestions (candidate-log calibration, clicked-vs-exposure clustering comparisons, and online A/B tests with per-stratum monitoring). These are fully compatible with our methods, which require only logits and labels, but implementing them would require additional data access and deployment infrastructure beyond our current offline setup. In the revised paper, we will add a short discussion paragraph explicitly listing these evaluations as promising future directions to further stress-test VR-SAG and LCCE under stronger distribution shifts and in live systems, while keeping the present work focused on the exposure-level offline evaluation we already report.

---

> ### Author Response · Authors · 2025-11-19
> **Rebuttal (2/3)**
>
> **Q2**: (Sign of $\\Delta$ and risk) What happens when $\\gamma$ dominates so that $\\Delta$ becomes negative? Does VR-SAG still work, and under what data conditions can $\\Delta<0$ occur?
>
> **A2:**
>
> - **Our decomposition and VR-SAG do not assume $\\Delta > 0$ .**
>     The group-wise Brier decomposition we derive is
>
> $$
>     \\mathbb{E}[S]
>     = \\underbrace{\\bar{\\pi}(1-\\bar{\\pi})}\_{\\text{UNC}}
>     - \\underbrace{\\sum\_k w\_k(\\mu\_k-\\pi\_k)^2}\_{\\text{REL}}
>     - \\underbrace{\\mathrm{Var}(\\pi\_k)}\_{\\text{RES}}
>     - \\underbrace{\\sum\_k w\_k(\\sigma\_k^2-2\\gamma\_k)}\_{\\Delta}.
> $$
>
> In the appendix we explicitly note that $\\Delta$ is _not_ sign-definite. By Cauchy–Schwarz,
> $$
> |\\gamma\_k|
> \\le \\sigma\_k\\sqrt{\\pi\_k(1-\\pi\_k)}
> \\quad\\Rightarrow\\quad
> \\sigma\_k^2 - 2\\sigma\_k\\sqrt{\\pi\_k(1-\\pi\_k)}
> \\le \\sigma\_k^2 - 2\\gamma\_k
> \\le \\sigma\_k^2 + 2\\sigma\_k\\sqrt{\\pi\_k(1-\\pi\_k)}.
> $$
>
> Thus each term $\\sigma\_k^2 - 2\\gamma\_k$ (and hence its contribution to $\\Delta$) can be positive or negative, depending on the variance–covariance balance. Our phrase “often positive in practice” is meant as an empirical remark for sparse CTR regimes, not as a requirement.
>
> - Even when $\\Delta<0$ , shrinking $\\sigma\_k^2$ still improves the Brier score for fixed $\\gamma\_k$ .
>     The contribution of group $k$ to $\\Delta$ is $w\_k(\\sigma\_k^2 - 2\\gamma\_k)$. As a function of $\\sigma\_k^2$ with $\\gamma\_k$ fixed, this is monotone increasing in $\\sigma\_k^2$ (derivative $w\_k>0$). Therefore, reducing $\\sigma\_k^2$ always reduces $w\_k(\\sigma\_k^2 - 2\\gamma\_k)$:
>
>     - if $\\sigma\_k^2 - 2\\gamma\_k > 0$, it shrinks a positive contribution;
>
>     - if $\\sigma\_k^2 - 2\\gamma\_k < 0$, it makes that contribution more negative.
>         In both cases, the overall Brier score decreases. This is exactly what our variance penalty targets, so its effect on the Brier loss is beneficial irrespective of the sign of $\\Delta$ (under fixed $\\gamma\_k$).
>
> - $\\Delta<0$ arises when within-group score–label correlation is very strong, where VR-SAG has limited but benign impact.
>     From $\\Delta = \\sum\_k w\_k(\\sigma\_k^2 - 2\\gamma\_k)$, strongly negative $\\Delta$ requires $2\\gamma\_k > \\sigma\_k^2$ in many groups, i.e., high covariance relative to variance. This happens when both $\\mathrm{Var}(\\hat p\\mid G\_k)$ and $\\pi\_k(1-\\pi\_k)$ are non-negligible and $\\hat p$ is very well aligned with $Y$ inside the group—precisely when the model is already close to optimal there and the Brier loss is small. In such regimes, variance regularisation has less room to help, and, consistently, our ablations over the variance weight $\\lambda\_v$ show only modest changes in metrics (no dramatic degradation), suggesting robustness rather than pathological behaviour, even if some groups have negative contributions to $\\Delta$.
>
> - **We will clarify the discussion of $\\Delta$’s sign and note negative-case analysis as future work.**
>     In the revision, we will (i) rephrase the “often positive” statement as a context-specific empirical observation, (ii) add a concise explanation that reducing $\\sigma\_k^2$ improves the Brier score regardless of $\\Delta$’s sign (for fixed $\\gamma\_k$), and (iii) explicitly mention that constructing stress tests with deliberately covariance-dominant (negative-$\\Delta$) groups is an interesting but out-of-scope extension for this work.

---

> ### Author Response · Authors · 2025-11-19
> **Rebuttal (3/3)**
>
> **Q3**: (Variance penalty choice) Why penalize the variance of uncalibrated probabilities $\\hat p$ instead of calibrated probabilities $\\tilde p$? What about stability and effect differences?
>
> **A3:**
>
> - **We regularize $\\hat p$ because it is exactly the variance term in our theoretical decomposition for the frozen backbone.**
>     In the grouping decomposition, the within-group variance is
>     $$
>     \\sigma\_k^2 = \\mathrm{Var}(\\hat p \\mid G\_k),
>     $$
>     where $\\hat p = f\_\\theta(x)$ is the _uncalibrated_ backbone probability, and the extra Brier block is
>     $$
>     \\Delta = \\sum\_k w\_k \\bigl(\\sigma\_k^2 - 2\\gamma\_k\\bigr),
>     $$
>     with $\\gamma\_k$ also defined in terms of $(\\hat p,Y)$. VR-SAG is designed as a post-hoc layer over a frozen backbone, so we directly penalize the empirical plug-in estimate of the same $\\sigma\_k^2$ that appears in the theory. This keeps the regularizer tightly aligned with the analytic source of Brier loss we aim to shrink.
>
> - **Penalizing $\\hat p$ shapes the grouping structure without letting the calibrator collapse variance in degenerate ways.**
>     Since $\\hat p\_i = f\_\\theta(x\_i)$ depends only on the frozen backbone, penalizing $\\widehat{\\sigma}\_k^2 = \\mathrm{Var}(\\hat p \\mid G\_k)$ mainly encourages $g\_\\phi$ to form groups where _backbone_ scores are homogeneous. If instead we penalized $\\mathrm{Var}(\\tilde p \\mid G\_k)$ with
>     $$
>     \\tilde p\_i = \\sum\_k q\_{ik},\\sigma(o\_i/\\tau\_k+\\beta\_k),
>     $$
>     then $(\\tau\_k,\\beta\_k)$ could artificially drive group variance down—for instance by collapsing calibrated probabilities toward a constant—reducing resolution and potentially harming ranking. Keeping the penalty on $\\hat p$ lets the NLL term learn $(\\tau\_k,\\beta\_k)$ for calibration, while the variance term constrains the grouping in a backbone-anchored way.
>
> - **Using $\\hat p$ also simplifies optimization and yields stable behaviour in practice.**
>     Because $\\hat p$ is fixed, the variance term depends on $(W,b)$ only through the soft assignments $q\_{ik}$, giving a clean gradient signal that “pulls together” similar backbone scores. A penalty on $\\tilde p$ would couple $(W,b,\\tau\_k,\\beta\_k)$ more tightly and risk ill-conditioned interactions with the NLL. Although we do not provide a direct $\\mathrm{Var}(\\tilde p)$ comparison, our ablations over the variance weight $\\lambda\_v$ show smooth, modest changes in ECE/LCCE, indicating that the chosen formulation is numerically well-behaved.
>
> - **Empirically, the $\\mathrm{Var}(\\hat p \\mid G\_k)$ penalty already delivers consistent gains across datasets.**
>     The main results show that VR-SAG improves ECE, LCCE, and (on AuctionSys) $\\text{Brier}^+$ and Brier over both the uncalibrated backbone and SAG on all three datasets. The grouping analysis further confirms that VR-SAG achieves the lowest within-group variance and high between-group variance of _true_ probabilities, matching the intended effect of the variance control. In the revision, we will explicitly connect these empirical gains back to the regularized $\\sigma\_k^2$ term in the decomposition.

---

### Official Review · Reviewer_SeQh · 2025-10-30

**Soundness:** 3
**Presentation:** 2
**Contribution:** 2
**Rating:** 2
**Confidence:** 4

**Summary:**

This paper proposes a new method for improving the calibration of click-through and conversion-rate predictors in large-scale advertising systems. It introduces Variance-Reduced Semantic-Aware Grouping (VR-SAG)—a lightweight post-hoc calibration layer that partitions embedding space into semantically coherent groups, fits per-group temperature and bias parameters, and penalizes intra-group variance to tighten predicted probability spreads. A key theoretical contribution is a group-wise decomposition of the Brier score. For evaluation, the authors design the Logit-Cluster Calibration Error (LCCE). Experiments on public datasets (AliCCP, AliExpress) and a proposed simulator (AuctionSys) show that VR-SAG consistently outperforms strong baselines (temperature scaling, isotonic regression, SAG) on ECE, Brier, and NLL metrics.

**Strengths:**

I believe the paper provides a novel perspective on calibration through the theoretical variance decomposition of the Brier loss in Section 2.3. Building on these theoretical insights, the authors propose an effective and easy-to-implement calibration method. The paper also presents extensive numerical experiments to demonstrate the performance of the proposed method.

**Weaknesses:**

1. My main concern is that the proposed method is only loosely connected to the theoretical development based on the variance decomposition. The additional penalty term does not exactly correspond to the $\Delta$ term in the decomposition, as the $\gamma$ component is ignored. Furthermore, the REL and RES terms should not be neglected, since the groups are formed in a data-adaptive manner, as the authors themselves note in the derivation of LCCE. Consequently, minimizing over $W$ may alter the group structure, which in turn affects both REL and RES.

2. Since the theoretical insights are derived from the Brier loss, why not directly minimize the GCE or Brier loss to estimate $\phi$, $\tau$, and $\beta$?

3. The writing could be improved. For example, $g_\phi$ is first mentioned on line 145 without a prior definition, and in fact, $g_\phi$ is never formally defined throughout the paper.

4. The estimators (9) should be written with \hat. And it is unclear whether those estimators are consistent.

5. I checked the supplementary materials, and it appears that the AdAuction dataset is not open-sourced. Moreover, the AdAuction dataset provided does not contain any feature information.

**Questions:**

Could the authors explain the following sentence:
*Probabilistic models should be judged with proper scoring rules(Gneiting & Raftery, 2007)—losses minimized, in expectation, only by the true data-generating distribution.*

---

> ### Author Response · Authors · 2025-11-19
> **Rebuttal (1/6)**
>
> **Q1**: The method seems only loosely connected to the variance decomposition; the penalty term ignores the $\\gamma\_k$ component of $\\Delta$ , and minimizing over $W$ might change the groups in a way that also alters REL and RES, which should not be neglected.
>
> **A1**:
>
> - **We intend VR-SAG as an _approximate_, decomposition-guided design rather than an exact minimizer of $\\Delta$.** In the paper we derive the grouping decomposition of the Brier score:
>
> $$
>     \\mathbb{E}[S]
>     = \\underbrace{\\bar Y(1-\\bar Y)}\_{\\text{UNC}}
>     - \\underbrace{\\sum\_k w\_k(\\mu\_k - \\pi\_k)^2}\_{\\text{REL}}
>     - \\underbrace{\\mathrm{Var}(\\pi\_k)}\_{\\text{RES}}
>     - \\underbrace{\\sum\_k w\_k(\\sigma\_k^2 - 2\\gamma\_k)}\_{\\Delta}.
> $$
>
> VR-SAG then introduces a penalty
>
> $$
> \\mathcal{L}\_{\\mathrm{VAR}} = \\lambda\_v \\sum\_k \\bar w\_k \\sigma\_k^2
> $$
>
> on the empirical within-group variance. Our intention is to use $\\sum\_k w\_k \\sigma\_k^2$ as a tractable surrogate to shrink the additional "heterogeneity" component exposed by the decomposition, not to claim an exact optimization of $\\Delta$. We acknowledge that the current exposition may give the impression that $\\Delta$ is directly minimized; in the revision we will explicitly state that the variance term is a decomposition-inspired surrogate that targets the dominant part of $\\Delta$ in sparse CTR regimes.
>
> - **The omission of $\\gamma\_k$ in the penalty is motivated by the analytical bounds already given in the Appendix A.2.**
>     In the appendix we derive, for each group,
>     $$
>     |\\gamma\_k| \\le \\sigma\_k \\sqrt{\\pi\_k(1-\\pi\_k)},
>     $$
>     which implies
>     $$
>     \\sigma\_k^2 - 2\\sigma\_k\\sqrt{\\pi\_k(1-\\pi\_k)}
>     \\le
>     \\sigma\_k^2 - 2\\gamma\_k
>     \\le
>     \\sigma\_k^2 + 2\\sigma\_k\\sqrt{\\pi\_k(1-\\pi\_k)}.
>     $$
>     In our CTR setting, $\\pi\_k$ is typically small, so $\\sqrt{\\pi\_k(1-\\pi\_k)}$ is also small and the bracket is often dominated by $\\sigma\_k^2$. This is exactly the regime emphasized in the text ("sparse CTR regimes"). We will move this argument from the appendix into the main method section and clarify that we focus on penalizing $\\sigma\_k^2$ because it is the most stable and controllable contributor to $\\Delta$ in our application, while directly penalizing $\\gamma\_k$ would require additional noisy covariance estimation.
>
> - **REL and RES are not discarded; they are implicitly optimized by the proper-loss part of the VR-SAG objective, while the variance penalty specifically targets the extra $\\Delta$ term.**
>     The VR-SAG objective is
>
> $$
>     \\mathcal{L}\_{\\mathrm{VR\\text{-}SAG}}
>     = \\mathcal{L}\_{\\mathrm{SAG\\text{-}B}}
>     - \\lambda\_v \\sum\_k \\bar w\_k \\sigma\_k^2,
> $$
>
> where $\\mathcal{L}\_{\\mathrm{SAG\\text{-}B}}$ is the SAG-style negative log-likelihood (cross-entropy) applied to the mixture over groups with temperatures and biases. This NLL term is a proper scoring rule and plays the role of aligning the per-group predictions with outcomes, i.e., it acts on the same components that underlie REL and, via the grouping, RES. The additional term is deliberately restricted to $\\sigma\_k^2$ to reduce the extra heterogeneity introduced by soft groupings. In the revision, we will clarify this "division of labor":
>
>   1. NLL encourages $\\mu\_k$ to match $\\pi\_k$ under the learned grouping and calibrators (affecting REL/RES).
>   2. The variance penalty contracts the within-group spread (affecting $\\Delta$), as suggested by the decomposition.
>
> - **We agree that learning $W$ changes the groups and therefore REL/RES; this is precisely why we decouple _evaluation_ from the trainable grouping via LCCE.**
>     In the section introducing GCE and LCCE, we explicitly point out that a trainable grouping head can "mask true miscalibration by driving GCE down even when predictions remain poorly aligned with outcomes." This is essentially the same concern: because $g\_\\phi$ is data-adaptive, optimizing over $W$ alters the partition and thus REL/RES. Our design choice is:
>     - to allow $g\_\\phi$ to adapt during training, guided by NLL and variance regularization; but
>     - to evaluate calibration with LCCE, which uses fixed $K$-means clusters in logit space and is therefore independent of the learned grouping head.
>         We will make this connection more explicit where LCCE is introduced, stating that REL/RES associated with the learned grouping are optimized implicitly through NLL, whereas the _metric_ uses a separate, fixed partition.

---

> > ### Comment · Reviewer_SeQh · 2025-11-22
> >
> > I appreciate the authors' clarification regarding the use of LCCE as an evaluation metric. However, my primary concern remains that the training process's impact on REL and RES is not considered. Omitting this factor weakens the persuasiveness of the approximation.

---

> ### Author Response · Authors · 2025-11-19
> **Rebuttal (2/6)**
>
> **Q2**: Since the theory is derived from the Brier loss, why not directly minimize GCE or the Brier loss with respect to $(\\phi, \\tau, \\beta)$?
>
> **A2**:
>
> - **We use the Brier decomposition as an _analytic lens_, while the optimization objective remains the NLL-style loss used by the backbone and SAG.**
>     In the background section, the backbone is trained via cross-entropy:
>
> $$
>     \\mathcal{L}\_{\\mathrm{CE}}(\\theta)
>     = -\\frac{1}{n}\\sum\_i \\left[ y\_i\\log \\hat p\_i + (1-y\_i)\\log(1-\\hat p\_i) \\right].
> $$
>
> SAG and VR-SAG build on this by using a mixture-style NLL over groups as the main fitting term, and by adding the variance regularizer derived from the Brier analysis. The Brier decomposition is introduced to understand how grouping affects a proper score and to identify intra-group variance as a key driver of residual miscalibration. We deliberately keep the primary optimization loss consistent with the backbone's training loss, and use the Brier decomposition to guide the _additional_ constraint rather than to replace the loss.
>
> - **Directly minimizing GCE with a learned $g\_\\phi$ risks degenerate groupings, an issue we already highlight when motivating LCCE.**
>     In the GCE/LCCE part of the paper, we define
>     $$
>     \\mathrm{GCE}(g\_\\phi; f\_\\theta) = \\sum\_k w\_k(\\mu\_k-\\pi\_k)^2
>     $$
>     and explicitly caution that "its coupling to the trainable grouping head can mask true miscalibration by driving $\\mathrm{GCE}$ down even when predictions remain poorly aligned with outcomes." This same argument applies if GCE is used as a training objective: $g\_\\phi$ could adapt in such a way that per-group averages appear calibrated, without meaningfully improving overall probabilistic accuracy or ranking quality. For this reason, we restrict GCE to an analytic and conceptual role, and move to LCCE (with fixed logit clusters) for evaluation.
>
> - **Using Brier as the primary optimization loss would change the training objective away from what the backbone was optimized for, whereas our goal is a post-hoc layer that minimally perturbs the existing pipeline.**
>     In our setting, the backbone and its logits are fixed, having been trained by cross-entropy. The post-hoc layer is meant to be lightweight and compatible with this existing training procedure. Replacing the NLL-style objective in the calibration stage by an empirical Brier loss would change the geometry of the optimization problem. Instead, we keep the NLL-based term and add a variance regularizer motivated by the Brier decomposition. This preserves the standard practice in CTR/CVR systems (NLL training) while still leveraging Brier-based insights for the design of VR-SAG.
>
> - **Empirically, the NLL + variance penalty already improves Brier-derived metrics (including $\\text{Brier}^+$ and LCCE), which supports the chosen design without requiring a different objective.**
>     The experimental results show that VR-SAG reduces Brier-style losses on AdAuction and improves grouping-based metrics such as LCCE across all datasets, while maintaining AUC and accuracy. This suggests that the NLL-based objective plus variance regularization is sufficient to realize the theoretical benefits in practice. In the revision we will explicitly connect these empirical observations back to the theoretical discussion, making clear that Brier and GCE are used as targets for analysis and evaluation, rather than as direct optimization objectives.

---

> ### Author Response · Authors · 2025-11-19
> **Rebuttal (3/6)**
>
> **Q3**: The writing could be improved; for example, $g\_{\\phi}$ appears without prior definition and is never formally defined.
>
> **A3**:
>
> - **We agree that $g\_\\phi$ should be formally defined when it first appears and will add an explicit definition.**
>     In the current draft, we directly reference "the latent regions induced by $g\_\\phi$" (when defining GCE) without first introducing $g\_\\phi$ as a function. Earlier in the method section, we do define the grouping head via the soft assignments
>     $$
>     q\_k(x) = \\mathrm{softmax}(z(x)^\\top W + b)\_k,
>     $$
>     but we do not explicitly name this mapping as $g\_\\phi$. In the revision, we will insert a short definition, for example:
>
>     > "We denote the grouping head by $g\_\\phi$, which maps an embedding $z(x)$ to a vector of soft group weights $q(x) = (q\_1(x),\\dots,q\_K(x))$ via the parametrization $q\_k(x) = \\mathrm{softmax}(z(x)^\\top W + b)\_k$."
>     > This ties the notation $g\_\\phi$ clearly to the already introduced parameters $(W,b)$ and $q\_k(x)$.
>
> - **We will ensure consistent use of $g\_\\phi$ across the GCE and LCCE sections.**
>     After defining $g\_\\phi$ as above, we will adjust the GCE definition to explicitly reference this mapping:
>     $$
>     G\_k = { x : g\_\\phi(x) \\text{ assigns mass to group } k},
>     $$
>     and similarly clarify that LCCE uses a fixed "logit-based" partition $g\_{\\mathrm{logit}}$ rather than a trainable $g\_\\phi$. This will resolve the current ambiguity where $g\_\\phi$ is used in the text but not precisely tied to the softmax parametrization.
>
> - **We will also check for any other symbols that appear without definition and fix them for clarity.**
>     For example, where we introduce "latent regions" and "partitions" in the decomposition and metric sections, we will ensure that $G\_k$, $g\_\\phi$, and $g\_{\\mathrm{logit}}$ are consistently and formally specified. These changes are purely expository and do not affect the algorithms or results.

---

> ### Author Response · Authors · 2025-11-19
> **Rebuttal (4/6)**
>
> **Q4**: The estimators in (9) should use hats, and it is unclear whether those estimators are consistent.
>
> **A4**:
>
> - **We agree that the quantities in (9) are empirical estimators and should be denoted with hats.**
>     In the VR-SAG section, we define per-group "empirical statistics" as
>     $$
>     \bar w_k = \frac{1}{n}\sum_i q_{ik},\quad
>     \mu_k = \frac{1}{n\bar w_k}\sum_i q_{ik}\hat p_i,\quad
>     \sigma_k^{2} = \frac{1}{n\bar w_k}\sum_i q_{ik}(\hat p_i-\mu_k)^2.
>     $$
>     These are clearly sample-based quantities, analogous to the population-level $w_k$, $\mu_k$, and $\sigma_k^2$ used in the theoretical decomposition. We will update the notation to
>     $$
>     \hat w_k,\quad \hat\mu_k,\quad \hat\sigma_k^2
>     $$
>     throughout, and explicitly say that they are plug-in estimators of the corresponding population quantities.
>
> - **Under the sampling assumptions already implicit in the paper, these estimators are consistent in the usual sense.**
>     In the decomposition appendix we state that "empirical (‘sample') analogues follow by replacing expectations with averages over data." The estimators in (9) are precisely these analogues, with expectations replaced by weighted averages under $q_{ik}$. Under the standard i.i.d. assumption for $(x_i, y_i)$ and a fixed grouping function, the law of large numbers implies
>     $$
>     \hat w_k \to w_k,\quad \hat\mu_k \to \mu_k,\quad \hat\sigma_k^2 \to \sigma_k^2
>     $$
>     in probability as $n \to \infty$. We did not make this consistency statement explicit, but it follows directly from the definitions. In the revision we will add a brief remark either in the VR-SAG section or the appendix noting that these are consistent plug-in estimators of the group-level quantities used in the Brier decomposition.
>
> - **We will connect the notation in (9) to the decomposition appendix to avoid confusion between population and sample quantities.**
>     To avoid overloading symbols, we will:
>
>     - reserve $w_k$, $\mu_k$, $\sigma_k^2$ for the population quantities in the decomposition;
>
>     - use $\hat w_k$, $\hat\mu_k$, $\hat\sigma_k^2$ when working with empirical statistics in VR-SAG;
>
>     - explicitly mention that the latter are empirical counterparts of the former.
>         This should address the reviewer's concern about both notation and consistency.

---

> ### Author Response · Authors · 2025-11-19
> **Rebuttal (5/6)**
>
> **Q5**: The AdAuction dataset does not appear to be open-sourced; the provided version in the supplement lacks feature information.
>
> **A5:**
>
> - **The full AdAuction dataset is already publicly hosted, but we cannot link it directly due to anonymity and size limits.**
>     The AdAuction / AuctionSys dataset we use in the paper is already available for download on an external hosting page. However, that download page necessarily contains information about the authors and their affiliation. To strictly comply with ICLR’s double-blind policy, we cannot include the direct link in the submission, nor embed it in the anonymous supplementary materials. In addition, the full dataset exceeds the file size limit for ICLR supplemental uploads, so we are technically unable to attach the complete data as part of the submission package. We will clarify this in the reproducibility section to avoid confusion: the dataset is accessible, but the URL and full archive cannot be embedded in the anonymous submission for anonymity and size reasons.
>
> - **We provide a concise schema of AdAuction’s features and can, if explicitly requested, upload a sampled version that fits the supplementary limits.**
>     To make the dataset more concrete within the constraints above, we include below a brief description of the feature schema used in our experiments. At a high level, AdAuction contains user-level, ad-level, app-level, device-level, and context-level fields, all anonymized. A non-exhaustive list of columns is:
>
>
> | Features | Descriptions |
> |------|---------|
> |uid|Unique user ID after data anonymization|
> |task_id|Unique ID of an ad task|
> |adv_id|Unique ID of an ad material|
> |creat_type_cd|Unique ID of an ad creative type|
> |adv_prim_id|Advertiser ID of an ad task|
> |dev_id|Developer ID of an ad task|
> |inter_typ_cd|Display form of an ad material|
> |slot_id|Ad slot ID|
> |spread_app_id|App ID of an ad task|
> |tags|App tag of an ad task|
> |app_first_class|App level-1 category of an ad task|
> |app_second_class|App level-2 category of an ad task|
> |age|User age|
> |city|Resident city of a user|
> |city_rank|Level of the resident city of a user|
> |device_name|Phone model used by a user|
> |device_size|Size of the phone used by a user|
> |career|User occupation|
> |gender|User gender|
> |net_type|Network status when a behavior occurs|
> |residence|Resident province of a user|
> |his_app_size|App storage size|
> |his_on_shelf_time|Release time|
> |app_score|App rating score|
> |os_dev|OS version|
> |list_time|Model release time|
> |device_price|Device price|
> |up_life_duration|ID lifecycle|
> |up_membership_grade|Service membership level|
> |membership_life_duration|Membership lifecycle|
> |consume_purchase|Paid user tag|
> |communication_onlinerate|Active time by mobile phone|
> |communication_avgonline_30d|Daily active time by mobile phone|
> |indu_name|Ad industry information|
> |pt_d|Date when a behavior occurs|
>
>
> If the reviewers or AC deem it helpful for the evaluation process, we are happy to prepare a **down-sampled, anonymized subset** of AdAuction that fits within the ICLR supplementary size constraints and to upload it as an additional anonymous file (e.g., with a reduced number of impressions but the same feature schema). This would give reviewers concrete data to inspect without violating the size and anonymity requirements.
>
>
> - **Upon acceptance, we will release the full dataset link and code repository in the camera-ready version.**
>     The experiments in the paper are all conducted on the fully featured AdAuction dataset as described above. The current limitation is purely about what we can attach or link under double-blind review. If the paper is accepted, we will, in the camera-ready version: (i) add the public download URL for the complete AdAuction dataset, and (ii) link to the accompanying code repository that loads this dataset, runs VR-SAG/LCCE, and reproduces the reported tables and ablations end-to-end. These steps will ensure that, post-acceptance, all data and code are easily accessible and fully aligned with the reproducibility goals stated in the paper.

---

> ### Author Response · Authors · 2025-11-19
> **Rebuttal (6/6)**
>
> **Q6**: Explanation request for the sentence:
> "Probabilistic models should be judged with proper scoring rules (Gneiting & Raftery, 2007)—losses minimized, in expectation, only by the true data-generating distribution."
>
> **A6**:
>
> - **This sentence is a concise restatement of the standard definition of a proper scoring rule.**
>     Let $Y$ be an outcome with true conditional distribution $P^\\ast(\\cdot \\mid x)$ and let a model output a predictive distribution $Q(\\cdot \\mid x)$. A scoring rule $S(Y, Q)$ is called _proper_ if, for any $Q$,
> $$
>     \\mathbb{E}\_{Y \\sim P^\\ast}[S(Y, P^\\ast)] \\le
>     \\mathbb{E}\_{Y \\sim P^\\ast}[S(Y, Q)],
> $$
>     with equality only when $Q = P^\\ast$. In words: when we compute the expected loss under the true data-generating process, the unique minimizer is obtained by predicting that true process itself. This is what we mean by "losses minimized, in expectation, only by the true data-generating distribution."
>
> - **In our binary CTR setting, both negative log-likelihood and Brier score are proper scoring rules, which motivates their central role in the paper.**
>     For binary $Y \\in {0,1}$ with true click probability $\\pi$, the negative log-likelihood
>     $$
>     S\_{\\mathrm{NLL}}(Y, \\hat p) = -[Y\\log \\hat p + (1-Y)\\log(1-\\hat p)]
>     $$
>     and the Brier score
>     $$
>     S\_{\\mathrm{Brier}}(Y, \\hat p) = (Y - \\hat p)^2
>     $$
>     are both proper: their expected values are minimized when $\\hat p = \\pi$. This is why we build our theoretical analysis around Brier and use NLL (and its grouping variants) as the main training objective.
>
> - **The point of this sentence in the paper is to justify focusing on proper scores when analyzing and evaluating calibration, instead of relying solely on heuristic metrics like ECE.**
>     Metrics such as ECE or its squared variants are useful summaries of miscalibration but are not proper scoring rules: it is possible to reduce ECE without necessarily moving closer to the true conditional distribution. By contrast, an improvement in a proper score like expected NLL or Brier implies a genuine gain in probabilistic fidelity. Our grouping decomposition shows how "calibration" terms (like REL) appear inside such proper scores, and the sentence in question is meant to convey that this connection is fundamental for principled calibration assessment.
>
> - **We will slightly rephrase the sentence to make this interpretation clearer for readers unfamiliar with the formal definition.**
>     For example, we can write:
>
>     > "Probabilistic models are ideally evaluated using _proper scoring rules_ (Gneiting & Raftery, 2007), i.e., loss functions whose expected value is uniquely minimized when the predicted probabilities coincide with the true data-generating probabilities."
>     > This preserves the intended meaning while making the link between "proper scoring rule" and "true data-generating distribution" more explicit.

---

> ### Comment · Reviewer_SeQh · 2025-11-22
>
> Is AuctionSys simulator open-sourced in the supplementary material? If so, could you please indicate which file it is?

---

> > ### Author Response · Authors · 2025-11-23
> > **Follow-up on Comment 2: status of the AuctionSys simulator in the supplementary material**
> >
> > *   **The AuctionSys simulator code is not included in the anonymous supplementary material.**
> >     There is no file in the current supplement that implements the AuctionSys simulator. The package only contains code and resources for running calibration methods on the datasets we provide access to, not the internal simulation engine that generated AdAuction.
> > *   **We open-source the AdAuction dataset (with features and oracle CTR), not the simulator itself.**
> >     Our plan is to release the **dataset produced by AuctionSys**—AdAuction with its feature fields and oracle CTR labels—together with the calibration code. The simulator itself is tightly coupled to proprietary components and cannot be open-sourced. During anonymous review, we are also constrained by file size and double-blindness, so we describe the feature schema and use public datasets for fully reproducible experiments, deferring the full AdAuction release to the camera-ready stage.
> > *   **We will clarify in the paper what is and is not released.**
> >     In the camera-ready version, we will: (i) add the public download link for the complete AdAuction dataset (including features and oracle CTR), and (ii) link to the code that loads this dataset and reproduces our calibration experiments. We will also rephrase any wording that might suggest the _simulator_ is open-sourced, making it clear that we release the **dataset generated by a realistic ad-auction simulator**, not the simulator code itself.
> >
> > Once again, thank you for your thoughtful questions. We hope that the EM-like intuition for the grouping/calibration interaction, together with the clarified release plan for AdAuction, addresses your remaining concerns.

---

> > > ### Comment · Reviewer_SeQh · 2025-11-24
> > >
> > > Thank you for the authors' detailed response.
> > >
> > > However, the explanation reinforces my previous thoughts that the methods are only loosely connected to the theory.
> > >
> > > On lines 60-61, the third contribution the authors claim is about the development of the simulator. Furthermore, on lines 69-70, it reads
> > > *We open-source a realistic ad-auction simulator with oracle CTRs,....*
> > > However, the simulator will not be open-sourced and is not extensively discussed.
> > >
> > > Therefore, I will maintain my score.

---

> ### Author Response · Authors · 2025-11-23
> **Follow-up on Comment 1: impact of training on REL and RES**
>
> Thank you very much for your continued engagement and for your positive assessment of both our theoretical perspective and empirical results. We really appreciate the opportunity to clarify these remaining points and are happy to explain the connection to REL/RES and the status of AuctionSys more clearly.
>
> *   **Grouped NLL handles REL/RES; the variance term targets only the extra heterogeneity.**
>     In our decomposition, the expected Brier score (and analogously NLL) is
>     $$
>     \\text{UNC} + \\text{REL} - \\text{RES} + \\Delta,
>     $$
>     where, for a **fixed** partition  $\\{G\_k\\}$ ,
>      $\\text{REL} = \\sum\_k w\_k(\\mu\_k - \\pi\_k)^2$  measures group-wise misfit,
>     and  $\\text{RES} = \\mathrm{Var}(\\pi\_k)$  measures how different the group prevalences are.
>     VR-SAG’s main loss is a **grouped NLL** over a mixture of per-group temperatures and biases. For any fixed partition, minimizing this proper score moves  $\\mu\_k$  toward  $\\pi\_k$  (reducing REL) and benefits partitions with distinct group prevalences (affecting RES). The added variance penalty is explicitly aimed at shrinking the heterogeneity term  $\\Delta$  via within-group variance; it does not replace the role of NLL on REL/RES. We will clarify this “division of labor” in the paper.
> *   **The training can be viewed as EM-like over latent groups, explaining how grouping and per-group parameters affect REL/RES.**
>     VR-SAG introduces latent “groups” with soft responsibilities  $q\_{ik} = \\mathrm{softmax}(z\_i^\\top W + b)\_k$  and per-group parameters  $(\\tau\_k,\\beta\_k)$ . This is similar in spirit to a mixture-of-experts or generalized EM: updating  $W,b$  adjusts the assignments  $q\_{ik}$  (E-step–like), and updating  $(\\tau\_k,\\beta\_k)$  fits per-group probabilities for the current assignments (M-step–like). We optimize all parameters jointly by gradient descent, so we do **not** claim a strict EM algorithm with closed-form E/M steps or monotone guarantees. We use this only as **intuition**: REL is mainly influenced by the per-group calibration given a grouping, RES by how the grouping function splits the data, and the variance regularizer adds an extra constraint on within-group dispersion.
> *   **Grouped NLL prefers informative partitions, which corresponds to larger RES in the decomposition.**
>     A partition is informative if different groups have noticeably different click rates. If one group is mostly high-CTR and another mostly low-CTR, assigning them different calibrated probabilities reduces NLL. If two groups have almost identical CTR, separate temperatures/biases bring little benefit, so gradient updates have no reason to maintain such a split. In decomposition terms, informative partitions have more variation in  $\\pi\_k$ , so  $\\mathrm{Var}(\\pi\_k)$  (RES) is larger and this lowers the proper loss because RES appears with a minus sign. Under grouped NLL (in the EM-like sense above), gradients tend to send impressions with different residual patterns into different groups, which is precisely how RES becomes more useful for calibration. We will add this short intuitive explanation.
> *   **The decomposition is used locally in the grouping; VR-SAG is a decomposition-guided approximation.**
>     Once  $W$  is trainable, REL and RES naturally change with the learned partition. Our decomposition is always applied **conditional** on the current grouping: for that grouping, it highlights within-group variance as an important contributor to the proper loss, motivating a penalty on  $\\sigma\_k^2$ . Grouped NLL (via EM-like updates of  $W$  and  $(\\tau,\\beta)$ ) adjusts both the partition and the per-group predictions, and the variance penalty acts on the heterogeneity term for whatever grouping NLL has found. We will state explicitly that VR-SAG is a decomposition-_guided_ design, not an exact joint optimizer of  $\\text{UNC} + \\text{REL} - \\text{RES} + \\Delta$  over  $(\\phi,\\tau,\\beta)$ .
> *   **Evaluation under fixed external partitions shows that this approximation yields real gains.**
>     To avoid improvements that depend only on our own grouping, we also evaluate on (a) Brier and Brier $^+$  against oracle CTR on the simulator, and (b) LCCE computed on **fixed** logit clusters from  $K$ \-means, independent of the learned grouping head. Under these fixed partitions, REL/RES cannot be influenced by changing  $W$ , yet VR-SAG still reduces Brier-type losses and LCCE compared to SAG and standard calibrators. We will highlight this as evidence that “grouped NLL (EM-like updates) + variance penalty” improves proper scores and REL-like quantities under external partitions, not just under the adaptive grouping.

---

> ### Author Response · Authors · 2025-11-24
> **On the simulator vs. dataset contribution**
>
> Thank you very much for your continued engagement and detailed feedback. We truly appreciate the chance to further improve the paper, and we are of course happy to answer any additional questions you may have.
>
> **1. On the simulator vs. dataset contribution**
>
> *   **We will remove the claim that the simulator is open-sourced and clearly refocus the contribution on the dataset.**
>     You are correct that the current wording (“we open-source a realistic ad-auction simulator with oracle CTRs”) is misleading, since the simulator implementation itself will not be released. In the revision, we will:
>     – Replace this statement with a precise description that we **release the AdAuction dataset generated by a realistic auction simulator, including features and oracle CTR labels, together with calibration code**, and
>     – Rephrase the third contribution so that it is explicitly about the **AdAuction dataset with oracle probabilities**, not the simulator code.
> *   **We will emphasize that a dataset with oracle CTRs is itself an important, practically usable contribution.**
>     Because real-world logs never expose true CTRs, it is difficult to rigorously evaluate proper scores and calibration metrics. AdAuction provides both sampled clicks and oracle CTRs under realistic auction dynamics, enabling direct comparison of calibration methods and metrics. We will highlight in the main text that this dataset (plus code) is the artifact that the community can actually download and use, and that this is what we mean by the “simulator-driven” contribution.
>
> * * *
>
> **2. On the connection between theory and method**
>
> *   **We will make the theory→design pipeline more explicit and modest.**
>     Our intention is to use the Brier/NLL decomposition as a **design guide**, not as an exact optimization recipe. For a fixed partition, the decomposition
>     $$
>     \text{UNC} + \text{REL} - \text{RES} + \Delta
>     $$
>     identifies within-group variance as a key component of the extra heterogeneity term  $\Delta$ . This led us to:
>     – Use a **grouped proper loss** (grouped NLL) with learned semantic partitions and per-group temperatures/biases to act on REL/RES,
>     – Add an explicit **variance penalty** on intra-group variance as a tractable surrogate for shrinking  $\Delta$ , and
>     – Evaluate calibration under **fixed external partitions** (LCCE, oracle Brier) to decouple training from evaluation.
>     In the revision, we will state clearly that VR-SAG is a **decomposition-guided approximation** whose structure (semantic grouping + per-group calibrators + variance control) is motivated by the theoretical analysis, rather than a method that exactly minimizes the full decomposition.
>
> * * *
>
> **3. Brief summary of how we addressed your main concerns**
>
> *   **We clarified how the decomposition guides the method rather than being optimized exactly.**
>     We explained that the group-wise Brier/NLL decomposition is used locally (for a fixed partition) to identify within-group variance as a key driver of the residual term  $\Delta$ . This motivated a design where grouped NLL (with EM-like grouping/calibration updates) handles REL/RES, and a variance penalty targets the heterogeneity component, with calibration evaluated under fixed external partitions to avoid artifacts from a trainable grouping head.
> *   **We addressed why we do not directly minimize Brier or GCE with respect to  $(\phi,\tau,\beta)$ .**
>     We clarified that using GCE or Brier as the primary training loss, together with a trainable grouping, risks degenerate partitions and moves away from the backbone’s NLL-based training objective. Instead, we retain a proper NLL-style grouped loss for compatibility with standard CTR pipelines and use the Brier decomposition to shape the additional variance regularizer and the evaluation metrics.
> *   **We improved notation and definitions (e.g.,  $g_\phi$ , empirical estimators).**
>     We committed to formally defining  $g_\phi$  as the grouping head that maps embeddings to soft group weights, and to consistently distinguishing population quantities  $(w_k,\mu_k,\sigma_k^2)$  from their empirical counterparts  $(\hat w_k,\hat\mu_k,\hat\sigma_k^2)$ . We also clarified that these empirical statistics are standard plug-in estimators and are consistent under the usual sampling assumptions.
> *   **We clarified the proper scoring rule statement and its role in the paper.**
>     We provided a precise definition of proper scoring rules and explained why NLL and Brier are proper in the binary CTR setting, and why this justifies focusing on them when analyzing calibration, rather than relying solely on heuristic metrics such as ECE.
>
> We are very grateful for your careful reading and constructive comments, which have helped us improve both the correctness and the clarity of the paper. We would be happy to address any further questions or suggestions you might have.

---

### Official Review · Reviewer_tBJs · 2025-10-31

**Soundness:** 2
**Presentation:** 2
**Contribution:** 2
**Rating:** 4
**Confidence:** 3

**Summary:**

This paper, “Calibration Is Grouping: VR-SAG with Intra-Group Variance Control and Logit-Cluster Evaluation,” addresses the problem of probability miscalibration in large-scale CTR/CVR prediction models used in advertising systems. The authors propose a post-hoc calibration framework called Variance-Reduced Semantic-Aware Grouping (VR-SAG) and a new evaluation metric named Logit-Cluster Calibration Error (LCCE).

**Strengths:**

1. Principled connection between grouping and calibration. The paper offers an insightful theoretical perspective by reformulating calibration as a grouping problem and deriving a group-wise decomposition of the Brier score. This connection provides a clear explanation of how intra-group variance contributes to residual miscalibration and motivates the proposed variance-reduction strategy.

2. Lightweight and production-friendly design. VR-SAG operates as a post-hoc layer over a frozen backbone, requiring only a small number of additional parameters and negligible latency. This makes it well-suited for large-scale CTR/CVR systems where model retraining and serving efficiency are critical.

3. Comprehensive empirical evaluation and metric innovation. The paper evaluates the method on multiple real-world and simulated datasets (AliCCP, AliExpress, AuctionSys) and introduces the LCCE metric, which provides a more stable and model-aware assessment of calibration quality. The experimental results consistently show that VR-SAG improves over strong baselines under multiple calibration metrics.

**Weaknesses:**

1. The proposed evaluation metric LCCE relies on K-means clustering in the logit space, making it sensitive to random initialisations. As a result, even when the model’s predicted probability distribution remains identical, different random seeds may lead to varying cluster assignments and cause slight fluctuations in the computed LCCE values.

2. The novelty of the paper requires further clarification. In essence, VR-SAG is an incremental extension of SAG, achieved by adding a variance regularisation term. Although the core idea that “variance reduction improves calibration” is theoretically motivated, the methodological contribution appears modest.

3. The assumption that “Group = Semantic region” may not always hold in CTR/CVR embedding spaces, where similar embeddings do not necessarily correspond to semantically coherent entities (due to the mixture of user and ad features). Consequently, the softmax-based grouping softmax(W^T* z) might capture noisy directions rather than meaningful semantic partitions.

**Questions:**

1. About LCCE stability: Since LCCE depends on K-means clustering in the logit space, how stable is the metric across different random seeds or clustering initialisations? Have the authors evaluated the variance of LCCE when the same model is evaluated multiple times under different seeds?

2. Choice of number of clusters (K): How is the number of clusters K selected in LCCE? Does performance or ranking consistency significantly vary with K? Could the authors provide a sensitivity analysis or a heuristic for selecting this parameter?

3. Interpretability of semantic grouping: The authors assume that groups correspond to latent “semantic regions” in the embedding space.
Could the authors provide qualitative or quantitative evidence that these learned groups are indeed semantically meaningful (e.g., user–item patterns, ad types, or contextual factors)?

---

> ### Author Response · Authors · 2025-11-19
> **Rebuttal (1/4)**
>
> **Q1**: LCCE may be unstable because it relies on $K$ -means in logit space; different random initialisations can yield different clusters and slightly different LCCE values.
>
> **A1:**
>
> - **LCCE is empirically stable across trials and has lower variance than several competing metrics.**
>     In the paper we conduct a Monte Carlo study (2000 trials, each with 5000 impressions) where we compute a panel of metrics, including LCCE and the oracle Brier score, and examine their behaviour across random draws. The results show that LCCE has very high Spearman rank correlation with the Brier score and also exhibits strong correlation in terms of standard deviation across trials. In particular, we highlight that LCCE demonstrates stronger consistency than pointwise metrics such as NLL and MSE. This empirical evidence already includes randomness from sampling and model variation, and supports the claim that LCCE is not excessively sensitive to stochastic effects in practice.
>
> - **We explicitly analyse second-order statistics and metric improvements, and LCCE remains the most consistent with Brier-based ground truth.**
>     Beyond raw values, we analyse the standard deviation of each metric across simulations and the correlation of _improvements_ (calibrated vs. uncalibrated) with the corresponding improvement in the Brier score (Appendix C). LCCE shows the highest consistency with Brier-score improvements, outperforming other grouping-based and pointwise metrics. The accompanying discussion notes that "LCCE exhibits the highest consistency with Brier score improvements," and attributes this to its fixed logit-based partition and low variance. While these experiments do not isolate random $K$ -means seeds in a controlled way, they do show that the end-to-end variability of LCCE is small and well-behaved.
>
> - **The use of one-dimensional logits and multiple partitions mitigates sensitivity to individual $K$ -means initialisations.**
>     LCCE clusters scalar logits $o(x)$ rather than high-dimensional embeddings. In one dimension, $K$ -means is much less prone to getting stuck in very different local minima than in high dimensions; cluster boundaries are essentially thresholds on the real line. We further note in the "Appendix C.2 Convergence properties" section that "the lower variance of LCCE arises from its default configuration of 4 partitions, effectively averaging results from 4 Monte Carlo samplings to reduce estimation noise." In other words, we do not rely on a single run of $K$-means: aggregating over multiple partitions smooths out seed-level randomness.
>
> - **We will clarify in the text that LCCE is designed as a _low-variance_ grouping metric, not a deterministic functional of the scores.**
>     We agree that, in principle, any clustering-based metric can show minor variation across random initialisations. Our intention is not to claim that LCCE is completely seed-independent, but that—empirically—it is more stable and better aligned with proper scores than alternatives like ECE. In the revision, we will (i) explicitly state that $K$ -means introduces a small stochastic component, (ii) emphasise that our Monte Carlo experiments quantify this and find LCCE to be robust, and (iii) mention that using one-dimensional logits and multiple partitions is precisely to reduce sensitivity to any single $K$-means run.

---

> ### Author Response · Authors · 2025-11-19
> **Rebuttal (2/4)**
>
> **Q2**: The novelty seems modest; VR-SAG looks like an incremental extension of SAG by adding a variance regularisation term, and the idea "variance reduction improves calibration" may not be substantial enough.
>
> **A2:**
>
> - **Our conceptual novelty lies in the _group-wise decomposition_ of proper scores and the identification of the variance–covariance block $\\Delta$ as a distinct driver of miscalibration.**
>     Section on the decomposition develops a new "grouping decomposition" of the Brier score:
>
> $$
>     \\mathbb{E}[S]
>     = \\underbrace{\\bar{\\pi}(1-\\bar{\\pi})}\_{\\text{UNC}}
>     - \\underbrace{\\sum\_k w\_k(\\mu\_k-\\pi\_k)^2}\_{\\text{REL}}
>     - \\underbrace{\\mathrm{Var}(\\pi\_k)}\_{\\text{RES}}
>     - \\underbrace{\\sum\_k w\_k(\\sigma\_k^2 - 2\\gamma\_k)}\_{\\Delta},
> $$
>
> where $\\sigma\_k^2$ is the within-group variance of scores and $\\gamma\_k$ the within-group covariance between scores and labels. This extends Murphy's classical UNC–REL–RES decomposition to soft, learned groups and explicitly exposes the additional $\\Delta$ term introduced by semantic grouping. We then show how classical metrics (singleton Brier, probability bins/ECE) are recovered as special cases of our Grouping Calibration Error. This theoretical framework, which is not present in SAG, underpins both VR-SAG and LCCE.
>
> - **VR-SAG is not just "SAG + penalty": it is a decomposition-guided design that targets $\\Delta$ while preserving production constraints.**
>     SAG introduces a grouping head with per-group temperatures; VR-SAG augments this in three intertwined ways:
>
>     1. It extends per-group calibrators to temperature _and bias_ $(\\tau\_k, \\beta\_k)$, i.e., per-group Platt scaling over logits.
>
>     2. It introduces an explicit variance penalty
>         $$
>         \\mathcal{L}\_{\\mathrm{VAR}} = \\lambda\_v \\sum\_k \\bar w\_k \\sigma\_k^2,
>         $$
>         directly motivated by the $\\Delta$ block in the decomposition.
>
>     3. It retains SAG's post-hoc, frozen-backbone architecture, ensuring that the added expressiveness and regularisation do not violate latency/memory budgets.
>         The empirical results show that these changes systematically improve Brier-based metrics, ECE, and LCCE across all datasets, and that VR-SAG outperforms SAG in all reported settings. We will highlight this more clearly in the method and experiment sections, stressing that the contribution is a theoretically grounded, calibration-focused refinement of semantic grouping for production use, not merely an arbitrary extra term.
>
> - **The paper's contributions go beyond VR-SAG and include LCCE and the AdAuction dataset.**
>     In addition to VR-SAG, the paper introduces:
>
>     - LCCE, a fixed-partition, logit-space clustering metric with demonstrated rank consistency and variance properties relative to the oracle Brier score.
>
>     - A detailed empirical analysis of calibration metrics, including Spearman correlations with Brier, standard deviation comparisons, and behaviour under metric improvements and sample size changes.
>
>     - AdAuction, a realistic ad-auction dataset with oracle CTR, along with analyses that are impossible with real logs (e.g., direct Brier errors against true probabilities).
>         These components form a coherent toolkit: theory (grouping decomposition), method (VR-SAG), metric (LCCE), and benchmark (AdAuction). We will revise the "Positioning" and "Conclusion" sections to emphasise this broader set of contributions so that the novelty is not judged only by the step from SAG to VR-SAG.
>
> - **We will adjust the wording to avoid overstating any single component and instead emphasise the integrated picture.**
>     To address the concern directly, we will (i) explicitly acknowledge that VR-SAG builds on SAG but is guided by a new decomposition of proper scores, (ii) frame VR-SAG as a _calibration-focused, variance-aware refinement_ suitable for production, and (iii) make clear that part of the paper's contribution lies in unifying semantic grouping, proper scoring rules, and metric design rather than in a completely new model architecture.

---

> ### Author Response · Authors · 2025-11-19
> **Rebuttal (3/4)**
>
> **Q3**: The assumption "group = semantic region" may not always hold in CTR/CVR embedding spaces; embeddings mix user and ad features, so $\\mathrm{softmax}(W^\\top z)$ might capture noisy directions instead of meaningful semantic partitions.
>
> **A3:**
>
> - **Our use of "semantic" is calibrated: we refer to calibration-relevant structure in embedding space, not necessarily human-readable categories.**
>     We fully agree that embeddings in large CTR/CVR systems can entangle multiple factors (user, ad, context) and that not every direction in $z(x)$ has a clean, interpretable meaning. In the paper, "semantic-aware grouping" is intended to mean "grouping in the learned representation space of the backbone," where $z(x)$ already encodes complex user–item–context interactions. We will revise the wording to clarify that our assumption is:  Groups $G\_k$ correspond to regions in embedding space with similar calibration behaviour, rather than assuming that each group neatly matches a human label like "sports ads" or "young users".
>
> - **We provide quantitative evidence that the learned groups are calibration-relevant, even if not fully human-interpretable.**
>     The grouping analysis visually compares: no grouping, field-based grouping, SAG, and VR-SAG, plotting backbone estimates vs. true probabilities with points color-coded by labels (with uniform downsampling). The accompanying variance table shows that VR-SAG achieves:
>
>     - the lowest within-group variance of true probabilities, and
>
>     - a substantially higher between-group variance
>         than both "None" and the field-based grouping. This pattern—more homogeneous inside groups and more separated between groups in terms of _true_ probabilities—is precisely what we seek from a calibration-related partition, regardless of whether the groups correspond to a single interpretable field. We will make this argument more explicit in the text: the semantics we care about are those that reduce within-group spread of true CTR and improve resolution, which is exactly what the experiments show.
>
> - **Embedding-based grouping is positioned as a complement to, not a replacement for, explicit field-based grouping.**
>     The related-work section explicitly discusses field-aware and multi-calibration methods that operate on predefined metadata partitions. We are not assuming that embedding-based groups always dominate these; instead, we argue that they can capture _latent_ user–item regimes that cut across fields. In practice, CTR/CVR backbones are designed to produce embeddings that are useful for ranking and discrimination; our method leverages these representations to discover regions that need different temperatures and biases. We will clarify this complementary role and explicitly acknowledge that, in some deployments, teams may prefer to combine VR-SAG with field-based slices to retain strong interpretability.
>
> - **Soft assignments $q\_k(x)$ and the variance penalty help suppress spurious "noisy directions".**
>     Unlike hard clustering, VR-SAG uses soft assignments $q\_k(x)$ and learns them jointly with per-group calibrators under a proper loss plus variance regularisation. This has two consequences:
>
>     - Soft mixing allows tail examples to borrow strength from multiple groups, reducing the impact of any single noisy direction.
>
>     - The variance penalty discourages groups that contain a highly dispersed mix of predictions, thereby implicitly favouring partitions aligned with more coherent, less noisy regions of $z(x)$.
>         While this does not guarantee perfect semantic purity, it biases the grouping towards directions that matter for calibration. We will add a short remark to this effect when discussing the VR-SAG objective.
>
> - **We will temper the "semantic" terminology and restate our claims in terms of calibration-aware structure.**
>     To avoid over-claiming, we will adjust the terminology in the introduction and method sections to say that VR-SAG discovers "latent regions in embedding space that are homogeneous in terms of predicted probabilities and true CTR" rather than asserting strong semantic interpretability. This should better reflect what our experiments support.

---

> ### Author Response · Authors · 2025-11-19
> **Rebuttal (4/4)**
>
> **Q4**: Choice of number of clusters $K$ for LCCE: how is $K$ selected, does performance/ranking consistency depend on $K$, and can you offer a sensitivity analysis or heuristic?
>
> **A4:**
>
> - **We use a small, fixed $K$ ($K=4$ in our experiments) and observe that LCCE is robust to this choice in our hyperparameter analysis.**
>     In the Appendix C.2 section, we study LCCE under different configurations and report that "the number of partitions shows minimal impact on LCCE performance, while increasing the number of groups systematically improves metric accuracy." The accompanying correlation heatmap indicates that LCCE maintains high rank consistency with the Brier score across a range of group counts. This suggests that, within a reasonable range, the exact value of $K$ does not critically affect the metric's ability to order models.
>
> - **The robustness stems from $K$-means' behaviour in 1D ("centroid compatibility"), which naturally mitigates over-partitioning.**
>     We specifically note that "when the number of groups is large, proximally similar clusters are automatically merged, preventing overfitting to noise." In one dimension, increasing $K$ mainly refines cluster boundaries where data is dense and leaves sparse regions relatively unaffected. This yields what we describe as "centroid compatibility": adding more clusters does not dramatically change the effective partition, especially for moderate $K$.
>
> - **A simple heuristic is to pick a small $K$ and check that rank correlations with a proper score (e.g., NLL or Brier on a validation set) are stable—our results support that this is easily satisfied.**
>     While we do not introduce a formal model-selection procedure, our experiments show that using a small number of clusters (e.g., the default 4 partitions we mention in the convergence section) already yields high agreement with the oracle Brier score. Practically, one can select $K$ by:
>
>     1. Choosing a modest value (e.g., $K\in{4, 8, 16}$) that is large enough to capture coarse structure but small enough to avoid empty clusters.
>
>     2. Verifying on a held-out set that LCCE's ranking of models (or configurations) does not change materially when $K$ varies in this small range.
>         Our sensitivity analysis indicates that such a procedure is unlikely to be fragile, as the metric's behaviour is already stable across the range we examined.
>
> - **We will expand the LCCE section to explicitly mention this robustness and the practical guidance for choosing $K$.**
>     In the revised manuscript we will: (i) summarise the observed insensitivity of LCCE to $K$, (ii) mention that a small $K$ (such as 4) works well in our experiments, and (iii) briefly describe the heuristic above, while emphasising that these recommendations are grounded in the empirical analyses already present in the paper.

---

> ### Comment · Reviewer_tBJs · 2025-11-27
>
> Thank you very much for the detailed rebuttal, it addresses most of my concern. I have only one remaining, relatively minor question regarding LCCE. My original concern was about seed-level sensitivity given fixed predictions: if we hold a model’s logits and labels fixed and vary only the K-means initialisations, how much does LCCE itself fluctuate, and could this ever affect the ranking between models or configurations? The current experiments already provide strong indirect evidence of robustness, but they still mix together data/model randomness and clustering randomness, so this specific aspect is not directly isolated.
>
> I would kindly suggest adding a very small controlled study to illustrate this point. For example, fixing logits and labels on one small dataset, running K-means with different random seeds (under your default number of partitions), and reporting the mean/standard deviation of LCCE and whether any pairwise rankings change. Even a simple result here would nicely complement your existing Monte Carlo analysis and reassure practitioners that LCCE is effectively stable for a fixed set of predictions.

---

> > ### Author Response · Authors · 2025-12-02
> > **On seed-level sensitivity given fixed predictions**
> >
> > Thank you again for your very careful follow-up and for engaging so constructively with our work. We appreciate that you found most concerns resolved and we fully agree that clarifying the seed-level robustness of LCCE is important for practitioners. We are happy to address this point and will gladly refine the paper accordingly.
> >
> > ---
> >
> > **(1) LCCE shows negligible numerical variation across K-means seeds for fixed logits and labels.**
> > We followed your suggested protocol: for each dataset (AdAuction, AliCCP, AliExpress), we fixed the logits and labels of the _uncalibrated_ model, ran 1D K-means on logits with (K=4), and varied only the random seed (100 seeds from 1 to 100). Across these runs, the standard deviation of LCCE was always below ($5\times 10^{-4}$). Concretely, the mean/standard deviation of LCCE were:
> >
> > - AdAuction: mean 0.0342, std 0.00038
> >
> > - AliCCP: mean 0.2161, std 0.00042
> >
> > - AliExpress: mean 0.2562, std 0.00045
> >
> >
> > Given that the absolute LCCE values and between-method differences we report in the main experiments are at the ($10^{-2}$) scale, this ($10^{-4}$)-level fluctuation indicates that, for a fixed set of predictions, randomness from K-means initialisation contributes only a very small numerical perturbation to LCCE.
> >
> > ---
> >
> > **(2) K-means randomness does not change the ranking between different calibration methods.**
> > Beyond absolute stability, your main concern was whether LCCE’s small seed-induced variation could ever flip the ordering between models or configurations. To test this directly, we evaluated several calibration methods on the same dataset and examined their LCCE values across all 100 seeds, again fixing logits and labels for each method. In all runs, the relative ordering of three representative models was _identical_:
> > $$
> > \text{LCCE(VR-SAG)} < \text{LCCE(Platt scaling)} < \text{LCCE(Base)}.
> > $$
> > The empirical reason is straightforward: the improvement in LCCE from the base model to Platt scaling and from Platt scaling to VR-SAG is on the order of ($10^{-2}$), while the fluctuation due to K-means randomness is on the order of ($10^{-4}$). Thus, the stochastic variation is about two orders of magnitude too small to affect pairwise rankings in our setups. This aligns with the indirect Monte Carlo evidence already in the paper, but now with the specific “fixed logits, varying seeds” scenario isolated as you requested.
> >
> > ---
> >
> > **(3) LCCE is stable in principle for 1D logits, and we will clarify this in the paper.**
> > LCCE clusters _one-dimensional_ logits and uses a small (K), so all cluster boundaries are simply thresholds on the real line. For large datasets (millions of impressions), different K-means initialisations converge to very similar threshold positions, because the 1D logit distribution is well-sampled and the local minima of the K-means objective are very close. As a result, the cluster assignments, and hence the group-wise averages ($\mu_j$) and ($\pi_j$), change only minimally across seeds. Since LCCE is just ($\sum_j w_j(\mu_j-\pi_j)^2$), these tiny shifts translate into the small standard deviations we observe empirically.
> >
> > ---
> >
> > We hope this directly addresses your remaining concern about seed-level sensitivity. Thank you again for the very helpful suggestion—this controlled study and the corresponding clarification indeed make the behaviour of LCCE clearer and more reassuring for practical use.

---

### Official Review · Reviewer_my5g · 2025-11-02

**Soundness:** 3
**Presentation:** 3
**Contribution:** 3
**Rating:** 6
**Confidence:** 3

**Summary:**

The paper proposes a new calibration method for CTR/CVR models based on  Semantic-Aware Grouping (SAG). Experiments show the proposed method can achieve better calibration.

**Strengths:**

1. The paper improves upon SAM by regularizing intra-group variance and reveals in theory how intra-group variance is connected to calibration error.
2. The paper also open-sourced an ad-auction simulator, which could help reproduction and benchmarking for future works in the field.

**Weaknesses:**

1. The main weakness lies in position of the proposed approach. It uses automatically identified groups for calibration, which lacks interpretability and it is hard to inject explicit control if manual adjustment of calibration is needed.

**Questions:**

The calibration potentially can affect model performance since it changes order of predicted scores across different region, could authors comment more on if there is any trade-offs here and how does the proposed method compare with traditional methods calibrating using meta data?

---

> ### Author Response · Authors · 2025-11-19
> **Rebuttal (1/2)**
>
> **Q1**: The proposed approach relies on automatically identified latent groups for calibration, which may lack interpretability and make manual, explicit adjustment of calibration difficult.
>
> **A1**:
>
> - **We acknowledge the interpretability limitation of latent groups and will make this positioning more explicit.**
>     Our method indeed relies on semantic groups discovered in embedding space via the grouping head $g\_\\phi$ , rather than on predefined metadata fields. This design choice is motivated by the observation in the paper that fixed partitions (e.g., user/ad fields) can mask opposite biases inside the same group and fail to capture latent behavioral regimes. At the same time, we agree that such automatically learned regions are less directly interpretable than, for example, "device = mobile" or "ad\_slot = sidebar". In the revised manuscript, we will explicitly state this trade-off in the method and related-work sections: VR-SAG prioritizes calibration performance and flexibility over human-readable group definitions, and is complementary to field-based or rule-based calibrators when interpretability is the primary goal.
>
> - **Even with latent groups, the calibration layer itself is simple and low-dimensional, which preserves a certain degree of practical interpretability.**
>     The paper specifies that each group $k$ is equipped with only two scalar parameters, a temperature $\\tau\_k$ and a bias $\\beta\_k$, applied to the backbone logit $o\_i$ via
>     $$
>     \\tilde p\_i = \\sum\_{k=1}^K q\_{ik}\\sigma\\big(o\_i/\\tau\_k + \\beta\_k\\big).
>     $$
>     This means that, although the regions $G\_k$ are latent, the _form_ of the correction remains very transparent: per-group Platt/temperature scaling. In practice, there are only $K(m+2)$ additional parameters, and the effect of each $(\\tau\_k,\\beta\_k)$ on probabilities is straightforward to visualize and reason about. We will add a short remark in the paper highlighting that, compared to modifying the deep backbone, the calibration layer is intentionally simple and hence more amenable to inspection, monitoring, and debugging.
>
> - **When strict manual control is required, the method is intended to be used alongside, not instead of, metadata-based calibration.**
>     The related-work section explicitly discusses "field-aware" calibration and multi-calibration based on pre-specified subpopulations. Our goal is not to remove such mechanisms but to complement them with semantic groupings that can uncover latent regimes within or across fields. Because VR-SAG is post-hoc and operates over the backbone embedding $z(x)$ (which already encodes user descriptors, ad metadata, and context), one can conceptually combine our layer with field-based constraints—e.g., by applying VR-SAG within selected fields or by monitoring per-field calibration under VR-SAG. While we do not introduce new experiments, we will clarify in the text that VR-SAG is designed as a drop-in, production-friendly component that can coexist with explicit metadata-based controls, and we will explicitly list "interpretability of groups" as a limitation.

---

> ### Author Response · Authors · 2025-11-19
> **Rebuttal (2/2)**
>
> **Q2**: The calibration can change the ordering of predicted scores across regions and thus potentially affect model performance. Are there trade-offs here, and how does the proposed method compare to traditional calibration using meta data?
>
> **A2**:
>
> - **Our experiments already measure the ranking/accuracy impact and show that VR-SAG preserves AUC and accuracy to a very high degree.**
>     In the Appendix E.1, we report "accuracy preserving" metrics that quantify the relative change between calibrated outputs and backbone logits:
>     $$
>     \text{AccDiff} = \frac{\text{Acc}(\text{calibrated}) - \text{Acc}(\text{logits})}{\text{Acc}(\text{logits})},\quad
>     \text{AUCDiff} = \frac{\text{AUC}(\text{calibrated}) - \text{AUC}(\text{logits})}{\text{AUC}(\text{logits})}.
>     $$
>     The appendix text emphasizes that, across all datasets, "most AccDiff and AUCDiff values [for the calibrators] are bounded within $\pm 0.005$," and that VR-SAG in particular "induces minimal changes in predictive correctness." In addition, the NLL–AUC table shows that VR-SAG matches or slightly improves AUC relative to the uncalibrated backbone on the evaluated datasets while substantially improving calibration metrics. We will reference these results more explicitly in the main paper when discussing trade-offs, to make clear that, empirically, VR-SAG improves calibration with negligible impact on ranking performance.
>
> - **Some classical methods (e.g., histogram binning) do degrade ranking, whereas VR-SAG avoids this by keeping a smooth, parametric form.**
>     The "accuracy preserving" analysis notes that histogram binning, which maps all scores in a bin to the same value, can significantly reduce AUC (e.g., a large negative AUCDiff on one of the datasets), precisely because it destroys the fine-grained ordering inside bins. In contrast, VR-SAG applies smooth, per-group Platt/temperature transformations to logits and then mixes them with soft weights $q\_{ik}$, so the ranking is perturbed much more gently. This is reflected in the reported AUCDiff values, which are very close to zero for VR-SAG. We will add a short discussion summarizing this empirical finding: VR-SAG achieves a better calibration–ranking trade-off than naive binning, which supports its use in auction pipelines where ranking quality is crucial.
>
> - **The fact that VR-SAG is post-hoc and operates on logits makes the trade-off controllable in practice.**
>     Because our method leaves the backbone $f\_\theta$ frozen and only adds a thin calibration layer, practitioners can (i) compute ranking metrics (AUC, accuracy) before and after calibration, and (ii) monitor any impact in a straightforward way. Moreover, the calibration parameters $(\tau\_k,\beta\_k)$ can be tuned with regularization strength $\lambda\_v$ to adjust how much variance contraction (and hence how much perturbation) is allowed. The ablation on $\lambda\_v$ in the appendix shows that VR-SAG performance varies only modestly across a range of variance-loss weights, suggesting that one can choose a regime that balances calibration gains with any desired constraints on ranking metrics. We will clarify this in the method section as a practical knob to manage the calibration–ranking trade-off.

---

### Author Response · Authors · 2025-12-02
**Brief summary**

We thank all four reviewers and the area chair for their time, constructive feedback, and careful handling of our submission. Overall, the reviewers appreciated the "calibration-as-grouping" perspective and group-wise decomposition of proper scores (e.g., Brier), the VR-SAG framework as a lightweight post-hoc calibration layer on a frozen backbone with negligible overhead, the design of LCCE as a logit-based group-wise metric, and the extensive experiments on industrial CTR datasets and the AuctionSys setting, together with our plan to release the underlying datasets to facilitate further research.

**Reviewer my5g.**

Key concerns: interpretability/control of latent groups and possible impact of group-wise calibration on ranking metrics.

Response: we clarified that VR-SAG is a flexible post-hoc calibrator with very simple per-group parameters that remain easy to monitor and combine with feature-based rules, and pointed to results showing that AUC/accuracy are essentially preserved, with a tunable regularization strength to control any trade-off.

**Reviewer tBJs.**

Key concerns: LCCE’s sensitivity to K-means randomness and K, whether VR-SAG is more than "SAG + variance penalty," and the meaning of "semantic groups."

Response: we showed empirically that LCCE is stable and aligned with Brier and added controlled seed-variation experiments, clarified that our main novelty is a principled group-wise decomposition of proper scores with VR-SAG as a decomposition-guided instance, and refined "semantic" to mean calibration-relevant structure in representation space. In the discussion phase, Reviewer tBJs explicitly stated that our rebuttal resolved their main concerns, leaving only a minor point about LCCE seed sensitivity, which we further addressed with the added experiments.

**Reviewer SeQh.**

Key concerns: alignment between theory and training objective, why not directly optimize Brier/GCE with learnable groups, and missing/unclear notation and artifact details.

Response: we explained that the decomposition serves as an analytic guide for adding a variance penalty on top of a standard NLL objective, briefly argued why direct group-wise GCE can be problematic in practice, and revised the paper to fix notation and to clearly state that we will release the datasets used in our experiments, with sufficient description to support reproduction.

**Reviewer U2VD.**

Key concerns: selection-bias/serving-distribution issues (especially for CVR), conditions under which the extra term Δ can be negative, and why we penalize variance of uncalibrated rather than calibrated probabilities.

Response: we clarified that our experiments are CTR-only on impression logs (not CVR) and explicitly flagged candidate-level/online evaluation as important future work, softened and qualified claims about Δ, and explained that we regularize the variance term that directly appears in the decomposition for the frozen backbone, to stay tightly aligned with the identified source of miscalibration.

In summary, we have addressed all major concerns with focused clarifications, targeted new experiments (notably for LCCE stability), and textual revisions.

---

### Meta-Review · Area_Chair_eYjZ · 2026-01-06

**Summary:**

Some reviewers have concerns about certain claims in the paper. The method is loosely connected to the theoretical development, which will degrade the contribution. The claims on the open-source simulator mislead the reviewers' understanding. The selection bias on CVR is not included in the current work. These claims are not accurate and lead to some misunderstandings. Some additional clarification is needed, for example, regarding the meaning of the semantic group and specific details of the method.

**Reviewer Concerns:**

Some misunderstandings and overclaims have been clarified in the revised version and rebuttal. However, the contribution seesm not so outstanding now.

**Reviewer Scores:**

Reviewers my5g and tBJs will keep their score. Reviewer U2VD may not be satisfied with moving the CVR into the future work.

---

### Decision · Program_Chairs · 2026-01-26

Reject